# Cortical network architecture for context processing in primate brain

**Zenas C Chao\*, Yasuo Nagasaka, Naotaka Fujii\***

Laboratory for Adaptive Intelligence, RIKEN Brain Science Institute, Wako-shi, Japan

**Abstract** Context is information linked to a situation that can guide behavior. In the brain, context is encoded by sensory processing and can later be retrieved from memory. How context is communicated within the cortical network in sensory and mnemonic forms is unknown due to the lack of methods for high-resolution, brain-wide neuronal recording and analysis. Here, we report the comprehensive architecture of a cortical network for context processing. Using hemisphere-wide, high-density electrocorticography, we measured large-scale neuronal activity from monkeys observing videos of agents interacting in situations with different contexts. We extracted five context-related network structures including a bottom-up network during encoding and, seconds later, cue-dependent retrieval of the same network with the opposite top-down connectivity. These findings show that context is represented in the cortical network as distributed communication structures with dynamic information flows. This study provides a general methodology for recording and analyzing cortical network neuronal communication during cognition.

**\*For correspondence:**
zenas.c.chao@gmail.com (ZCC);
na@brain.riken.jp (NF)

**Competing interests:** The authors declare that no competing interests exist.

## Introduction

Context is the contingent sensory or cognitive background for a given situation. Different contexts can dramatically alter perception, cognition, or emotional reactions and decision-making, and in the brain network context can be represented during sensory encoding or mnemonic retrieval. The study of context is important for understanding the link between perception and cognition, in terms of both behavioral and neural processing, and the neural mechanisms underlying contextual information processing have been studied in a variety of domains including visual perception (*Bar, 2004*; *Schwartz et al., 2007*), emotion (*De Gelder, 2006*; *Barrett et al., 2007*; *Barrett and Kensinger, 2010*), language (*Hagoort, 2005*; *Aravena et al., 2010*), and social cognition (*Ibañez and Manes, 2012*). In the brain, context is proposed to require an interplay between bottom-up and top-down information processing in distributed neural networks (*Tononi and Edelman, 1997*; *Friston, 2005*). However, a comprehensive functional view of the brain circuits that mediate contextual processing remains unknown because bottom-up and top-down processes are often concurrent and interdependent, making the temporal and spatial resolution of their neural network organization difficult to separate.

To understand contextual information processing, we developed a fundamentally new approach to study high-resolution brain network architecture. The approach combines broadband neural recording of brain activity at high spatial and temporal resolution with big data analytical techniques to enable the computational extraction of latent structure in functional network dynamics. We employed this novel methodological pipeline to identify functional network structures underlying fast, internal, concurrent, and interdependent cognitive processes during context processing in monkeys watching video clips with sequentially staged contextual scenarios. Each scenario contained a conspecific showing emotional responses preceded by different situational contexts. With specific combinations of context and response stimuli, this paradigm allowed an examination of context-dependent brain activity and behavior by isolating context processing as a single variable in the task.

**eLife digest** If we see someone looking frightened, the way we respond to the situation is influenced by other information, referred to as the 'context'. For example, if the person is frightened because another individual is shouting at them, we might try to intervene. However, if the person is watching a horror video we may decide that they don't need our help and leave them to it. Nevertheless, it is not clear how the brain processes the context of a situation to inform our response.

Here, Chao et al. developed a new method to study electrical activity across the whole of the brain and used it to study how monkeys process context in response to several different social situations. In the experiments, monkeys were shown video clips in which one monkey—known as the 'video monkey'—was threatened by a human or another monkey, or in which the video monkey is facing an empty wall (i.e., in three different contexts). Afterwards, the video monkey either displays a frightened expression or a neutral one. Chao et al. found that if the video monkey looked frightened by the context, the monkeys watching the video clip shifted their gaze to observe the apparent threat. How these monkeys shifted their gaze depended on the context, but this behavior was absent when the video monkey gave a neutral expression.

The experiments used an array of electrodes that covered a wide area of the monkeys' brains to record electrical activity of nerve cells as the monkeys watched the videos. Chao et al. investigated how brain regions communicated with each other in response to different contexts, and found that the information of contexts was presented in the interactions between distant brain regions. The monkeys' brains sent information from a region called the temporal cortex (which is involved in processing sensory and social information), to another region called the prefrontal cortex (which is involved in functions such as reasoning, attention, and memory). Seconds later, the flow of information was reversed as the monkeys utilized information about the context to guide their behavior.

Chao et al.'s findings reveal how information about the context of a situation is transmitted around the brain to inform a response. The next challenge is to experimentally manipulate the identified brain circuits to investigate if problems in context processing could lead to the inappropriate responses that contribute to schizophrenia, post-traumatic stress disorder and other psychiatric disorders.

To measure large-scale brain network dynamics with sufficient resolution, we used a 128-channel hemisphere-wide high-density electrocorticography (HD-ECoG) array to quantify neuronal interactions with high spatial, spectral, and temporal resolution. This ECoG system has wider spatial coverage than conventional ECoG and LFP (*Buschman and Miller, 2007*; *Pesaran et al., 2008*; *Haegens et al., 2011*) and single–unit activity (*Gregoriou et al., 2009*), higher spatial resolution than MEG (*Gross et al., 2004*; *Siegel et al., 2008*), broader bandwidth than EEG (*Hipp et al., 2011*), and superior temporal resolution to fMRI (*Rees and House, 2005*; *Freeman et al., 2011*). After recording, we interrogated large-scale functional network dynamics using a multivariate effective connectivity analysis to quantify information content and directional flow within the brain network (*Blinowska, 2011*; *Chao and Fujii, 2013*) followed by big data analytical approaches to search the database of broadband neuronal connectivity for a latent organization of network communication structures.

## Results

### Large-scale recording of brain activity during video presentations

Monkeys watched video clips of another monkey (video monkey, or vM) engaging with a second agent (*Figure 1*) while cortical activity was recorded with a 128-channel ECoG array covering nearly an entire cerebral hemisphere. Three monkeys participated, one with a right hemisphere array (Subject 1), and two in the right (Subjects 2 and 3) (*Figure 1—figure supplement 1*). The data are fully accessible online and can be downloaded from the website Neurotycho.org.

The video clips started with a context between the two agents (*Context* period) followed by a response to the context (*Response* period). Six different video clips were created from three

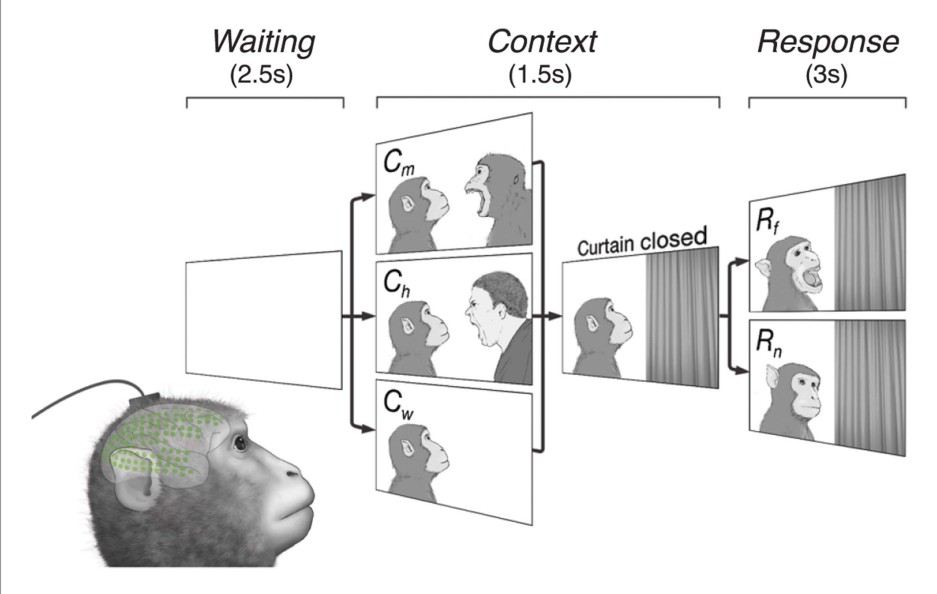

**Figure 1**. Subjects observe situational contexts with high-density electrocorticography (HD-ECoG) recording. We recorded 128-channel HD-ECoG signals from monkeys viewing video clips of a conspecific under three different situational contexts and two responses. The subject (lower-left, green circles represent ECoG electrodes) was seated in front of a TV monitor showing video clips consisting of a *Waiting* period of 2.5 s followed by a *Context* period of 1.5 s with one of three interactions between a video monkey (vM) on the left and a second agent on the right: vM threatened by a human ($C_h$), threatened by another monkey ($C_m$), or an empty wall ($C_w$). Next, a curtain closed to conceal the second agent followed by a *Response* period of 3 s with the vM showing either a frightened ($R_f$), or neutral expression ($R_n$). Pairwise combination of the contexts and responses produced six different video clips.

The following figure supplement is available for figure 1:

**Figure supplement 1**. Electrode locations in 3 subjects.

contexts, vM threatened by a human ($C_h$), threatened by another monkey ($C_m$), or facing an empty wall ($C_w$), combined with two responses, vM showing a frightened expression ($R_f$), or neutral expression ($R_n$), which were termed $C_hR_f$, $C_mR_f$, $C_wR_f$, $C_hR_n$, $C_mR_n$, and $C_wR_n$ (see *Videos 1–6*). Each video contained audio associated with the event, for example, sounds of a threatening human ($C_h$) and a frightened monkey ($R_f$). Each video represents a unique social context-response scenario, For example, $C_hR_f$ shows a human threatening a monkey (vM) followed by the monkey's frightened response. These staged presentations were designed to examine whether different contexts ($C_h$, $C_m$, or $C_w$) would give rise to context-dependent brain activity even with the same responses ($R_f$ or $R_n$).

## Eye movements demonstrate context- and response-dependent behaviors

During the task, subjects freely moved their eyes to observe the video interactions. We monitored eye movements to examine these spontaneous behavioral reactions and the associated zones in the video. We divided the trials into two conditions based on whether the context stimulus was visually perceived: $C^+$ where the subject was looking at the screen during the *Context* period, and $C^-$ where the subject was either closing its eyes or looking outside of the screen. Example eye movements are shown in *Figure 2—figure supplement 1*.

We first investigated which side of the video monitor the monkey attended. When the context was perceived ($C^+$) and the response stimulus was $R_f$, subjects focused more on the right section during the *Response* period, indicating interest in the curtain, or the threat behind the curtain, than the frightened vM (*Figure 2A*). This preference was absent when the response stimulus was $R_n$ or the context was not visually perceived ($C^-$). This is behavioral evidence that gaze direction preference

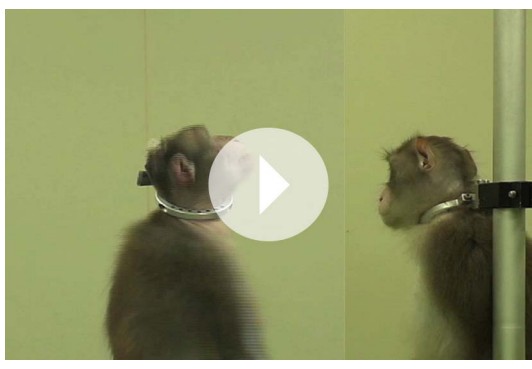

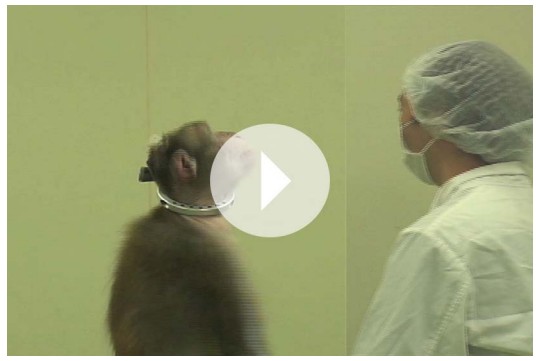

**Video 1.** Video clip for $C_mR_f$ trials. The clip contains the *Context* and *Response* periods.

**Video 2.** Video clip for $C_hR_f$ trials. The clip contains the *Context* and *Response* periods.

required not only the vM response, but also perception of the preceding context, which demonstrated a cognitive association between the perception of the context and response stimuli.

We then compared gazing behaviors from different trial types to identify behaviors selective to different scenarios ($C_hR_f$, $C_mR_f$, $C_wR_f$, $C_hR_n$, $C_mR_n$, or $C_wR_n$) and conditions ($C^+$ or $C^-$). We performed nine pairwise comparisons on gaze positions from different scenarios, separating $C^+$ and $C^-$ conditions, to examine their context and response dependence. For context dependence, we compared behaviors from trials with different context stimuli but the same response stimulus (6 comparisons: $C_mR_f$ vs $C_wR_f$, $C_wR_f$ vs $C_hR_f$, and $C_hR_f$ vs $C_mR_f$ for context dependence in $R_f$; $C_mR_n$ vs $C_wR_n$, $C_wR_n$ vs $C_hR_n$, and $C_hR_n$ vs $C_mR_n$ for context dependence in $R_n$). For response dependence, we compared behaviors from trials with the same context stimulus but with different response stimuli (3 comparisons: $C_mR_f$ vs $C_mR_n$, $C_wR_f$ vs $C_wR_n$, and $C_hR_f$ vs $C_hR_n$).

A context and response dependence was found in gazing behavior (*Figure 2B*). In $C^+$, significant differences in gaze position were found during the *Response* period between $C_mR_f$ and $C_wR_f$, and $C_wR_f$ and $C_hR_f$, but not between $C_hR_f$ and $C_mR_f$ (blue circles in left panel). This indicated that gaze shifting in $C_mR_f$ and $C_hR_f$ was comparable and stronger than in $C_wR_f$. This context dependence was absent when the response stimuli were $R_n$ (green circles in left panel). Furthermore, a significant response dependence was found during the *Response* period for all contexts ($C_mR_f$ vs $C_mR_n$, $C_wR_f$ vs $C_wR_n$, and $C_hR_f$ vs $C_hR_n$) (red circles in left panel) consistent with the results described in *Figure 2A*. In $C^-$, the context and response dependence found in $C^+$ was absent (right panel). These results indicated that the subjects' gaze shift during the *Response* period showed both response dependence ($R_f > R_n$) and context dependence ($C_m \approx C_h > C_w$), but only when context was perceived ($C^+ > C^-$).

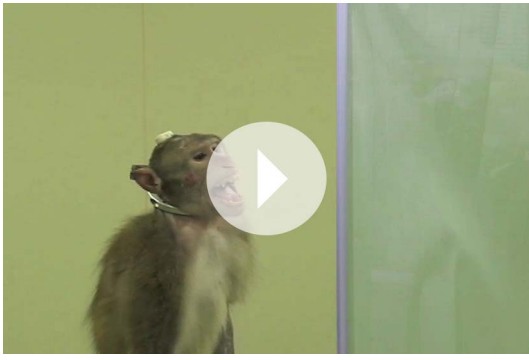

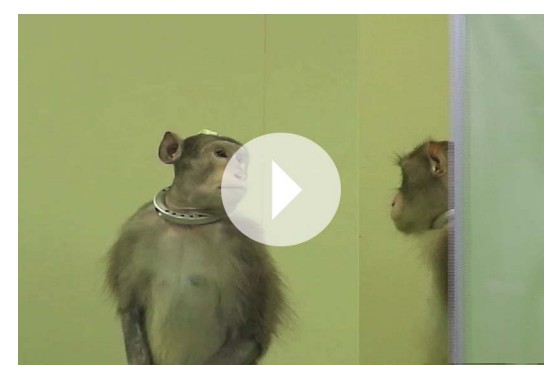

**Video 3.** Video clip for $C_wR_f$ trials. The clip contains the *Context* and *Response* periods.

**Video 4.** Video clip for $C_mR_n$ trials. The clip contains the *Context* and *Response* periods.

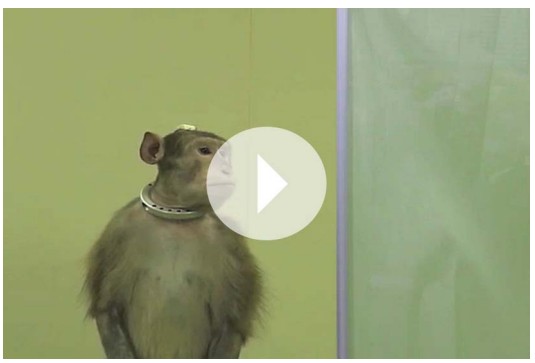

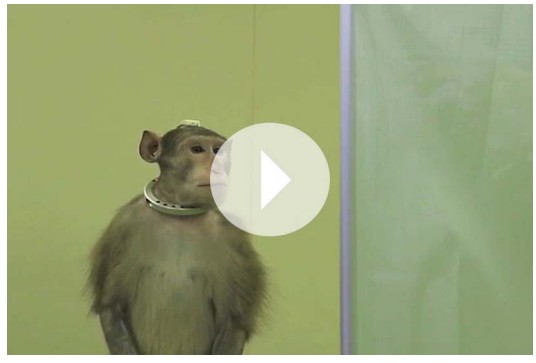

**Video 5.** Video clip for $C_hR_n$ trials. The clip contains the *Context* and *Response* periods.

**Video 6.** Video clip for $C_wR_n$ trials. The clip contains the *Context* and *Response* periods.

## Mining of large-scale ECoG data for cortical network interactions

To analyze the large-scale ECoG dataset, we identified cortical areas over the 128 electrodes in the array by independent component analysis (ICA). Each independent component (IC) represented a cortical area with statistically independent source signals (*Figure 3—figure supplement 1*, and experimental parameters in *Table 1*).

We then measured the causality of a connection from one cortical area (source area) to another (sink area) with a multivariate effective connectivity measure based on Granger causality: direct directed transfer function (dDTF) (*Korzeniewska et al., 2003*), which can represent phase differences between the two source signals to provide a time-frequency representation of their asymmetric causal dependence. We acquired dDTFs from all connections for each trial type (12 types: six scenarios and two conditions), and measured event-related causality (ERC), by normalizing the dDTF of each time point and each frequency bin to the median of the corresponding baseline control values. Thus, ERCs represent the spectro-temporal dynamics of network interactions evoked by different scenarios and conditions. Examples of ERCs are shown in *Figure 3A*.

We compared ERCs from different trial types to identify networks selectively activated in different scenarios ($C_hR_f$, $C_mR_f$, $C_wR_f$, $C_hR_n$, $C_mR_n$, or $C_wR_n$) and conditions ($C^+$ or $C^-$). We performed nine pairwise comparisons on ERCs from different scenarios, separating $C^+$ and $C^-$ trials, to examine their context and response dependence. To examine context dependence, we compared ERCs from trials with different contexts but the same response (6 comparisons). In contrast, to examine response dependence, we compared ERCs from trials with the same context but with different responses (3 comparisons). This approach is similar to the eye movement analysis (*Figure 2B*). The comparisons were performed with a subtractive approach to derive a significant difference in ERCs (ΔERCs, *Figure 3B*). Hence, ΔERCs revealed network connections, with corresponding time and frequency, where ERCs were significantly stronger or weaker in one scenario compared to another.

We pooled ΔERCs from all comparisons, conditions, connections, and subjects, to create a comprehensive broadband library of network dynamics for the entire study. To organize and visualize the dataset, we created a tensor with three dimensions: *Comparison-Condition*, *Time-Frequency*, and *Connection-Subject*, for the functional, dynamic, and anatomical aspects of the data, respectively (*Figure 2C*). The dimensionality of the tensor was 18 (nine comparisons under two conditions) by 3040 (160 time windows and 19 frequency bins) by 4668 (49 × 48 connections for Subject 1, 33 × 32 for Subject 2, and 36 × 35 for Subject 3).

To extract structured information from this high-volume dataset, we deconvolved the 3D tensor into multiple components by performing parallel factor analysis (PARAFAC), a generalization of principal component analysis (PCA) to higher order arrays (*Harshman and Lundy, 1994*) and measured the consistency of deconvolution under different iterations of PARAFAC (*Bro and Kiers, 2003*). Remarkably, we observed five dominant structures from the pooled ΔERCs that represented functional network dynamics, where each structure contained a comprehensive fingerprint of network function, dynamics, and anatomy (*Figure 3D*, and *Figure 3—figure supplement 2*). These five structures were robust against model order selection for ICA (*Figure 3—figure supplement 3*).

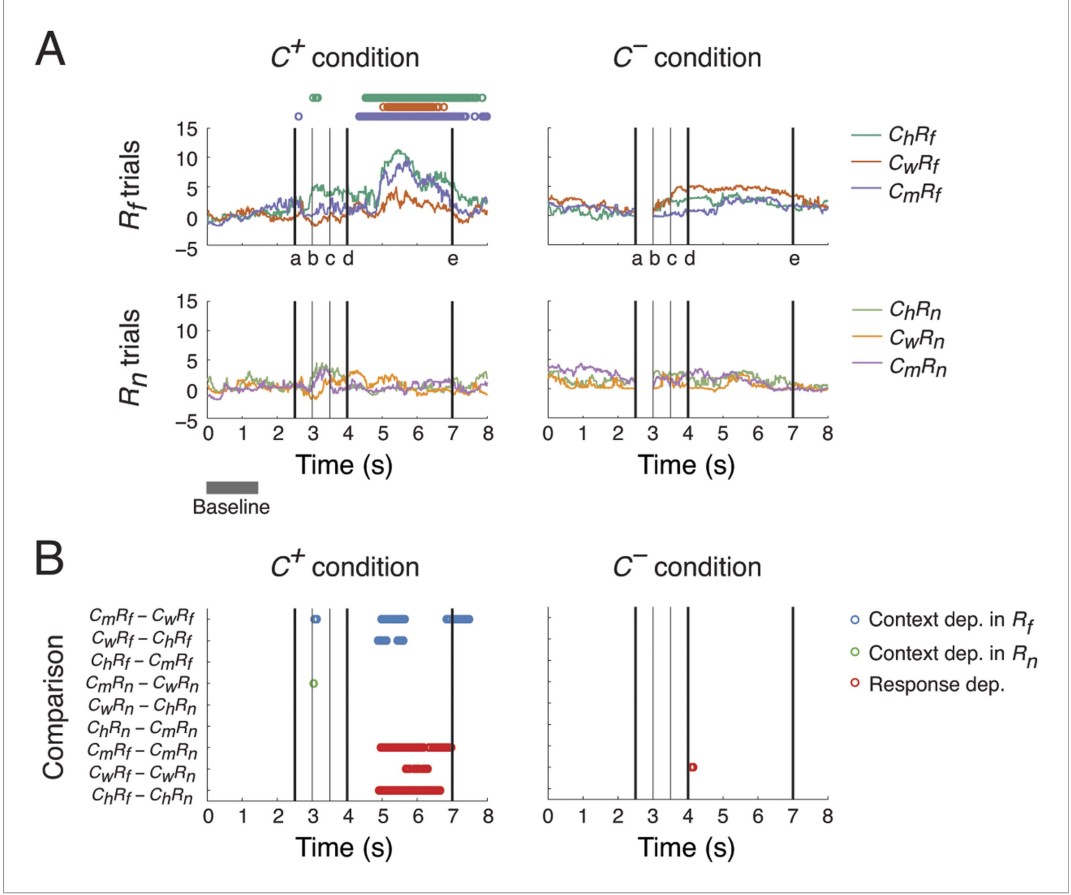

**Figure 2**. Context- and response-dependent eye movements. (**A**) Measurements of gaze shifting revealed a behavioral association between the context and response phases of the task. The gaze positions averaged from three subjects are shown for each trial type ($C_hR_f$, $C_mR_f$, $C_wR_f$, $C_hR_n$, $C_mR_n$, and $C_wR_n$) and condition ($C^+$ and $C^-$). Gaze shifting was quantified by gaze positions significantly different from baseline values ($\alpha_{Bonf} = 0.05$, baseline: gray bar), and was found only in $R_f$ trials under the $C^+$ condition (upper-left panel, the timing of gaze shifts are indicated on top, where the color represents the trial type indicated on the right). Black vertical lines represent the following events (see labels on the x-axis): (a) onset of the *Context* period, (b) the curtain starts closing, (c) the curtain is fully closed, (d) onset of the *Response* period, and (e) end of the *Response* period (onset of the next trial). (**B**) Context and response dependence in gazing behavior. Gaze positions between different trial types were compared, separately in $C^+$ and $C^-$. For each comparison (y-axis), the timing of significant differences are shown as circles ($\alpha_{Bonf} = 0.05$), where blue, green, and red circles represent context dependence in $R_f$, context dependence in $R_n$, and response dependence, respectively. Gazing behavior showed both response dependence and context dependence, but only in $C^+$.

The following figure supplement is available for figure 2:

**Figure supplement 1**. Examples of eye movement.

## Discrete structured representations of functional network dynamics

The five structures are shown in *Figure 4* (Structures 1 and 2) and *Figure 5* (Structures 3, 4, and 5). Each structure represented a unique functional network dynamics, described by its compositions in the three tensor dimensions. The first tensor dimension (panel A) represented the differences across comparisons for each structure. We identified the significant differences and reconstructed the activation levels to show how each structure was activated under different scenarios and conditions (see the 'Materials and methods'). The second tensor dimension (panel B) represented spectro-temporal dynamics for each structure. The third tensor dimension (panel C) represented the anatomical connectivity pattern for each structure. We measured three connectivity statistics: (1) *causal density* is the sum of all outgoing and incoming causality for each area, showing areas with busy interactions; (2) *causal outflow* is the net

**Table 1**. Experimental parameters

| | | Subject 1 | Subject 2 | Subject 3 |
|---|---|---|---|---|
| Experiment | Hemisphere implanted | Right | Left | Left |
| | # of electrodes | 128 | 128 | 128 |
| | # of trials per class | 150 | 150 | 150 |
| | # of trials preserved per class (mean ± std) (see trial screening in 'Materials and methods') | 117.7 ± 3.5 | 122.2 ± 3.1 | 109.5 ± 3.6 |
| | # of $C^+$ trials per class (mean ± std) | 64.8 ± 5.2 | 60.3 ± 6.7 | 57.3 ± 5.6 |
| | # of $C^-$ trials per class (mean ± std) | 52.8 ± 1.9 | 61.8 ± 8.1 | 52.2 ± 3.5 |
| ICA (see *Figure 3—figure supplement 1*) | # of ICs for 90% variance explained | 58 | 38 | 39 |
| | # of ICs preserved (see IC screening in 'Materials and methods') | 49 (removed ICs 1, 2, 3, 4, 5, 11, 44, 46, and 47) | 33 (removed ICs 1, 2, 7, 8, and 29) | 36 (removed ICs 2, 10, and 27) |

outgoing causality of each area, indicating the sources and sinks of interactions; and (3) *maximum flow between areas* is the maximal causality of all connections between cortical areas (7 areas found with busy interactions were chosen) (see results for individual subjects in *Figure 4—figure supplements 1–3*). The extracted statistics were robust across all subjects with different electrode placements suggesting that the structures were bilaterally symmetric across hemispheres (*Figure 4—figure supplement 4*).

Structure 1 was activated first, with context dependence only in the *Context* period ($C_m > C_h > C_w$) suggesting sensory processing that can discriminate between contextual stimuli, that is, context perception. The context dependence was weaker but remained in $C^-$ ($C^+ > C^-$), suggesting that auditory information in context stimuli was processed. The spectral dynamics of Structure 1 emerged primarily in the high-γ band (>70 Hz), and contained mostly bottom-up connections from the posterior to anterior parts of temporal cortex.

Structure 2 was the earliest activated in the *Response* period, with only response dependence ($R_f > R_n$), and was independent of whether context stimuli were visually perceived ($C^+ \approx C^-$). Thus, Structure 2 corresponds to sensory processing that can discriminate between response stimuli, that is, response perception. Spectrally, Structure 2 emerged in both high-γ and β bands, and contained connections similar to those in Structure 1, with an additional communication channel from anterior temporal cortex to the prefrontal cortex (PFC). Therefore, Structures 1 and 2 represent the multisensory processing of audiovisual stimuli, and Structure 2 could underlie the additional evaluation of emotional valence associated with the stimuli.

Structure 3 was activated the second earliest in the *Context* period showing a generalized context dependence ($C_m \approx C_h > C_w$) representing the abstract categorization of the context ('an indeterminate agent is threatening vM'). Similar to Structure 1, the dependence in Structure 3 was weaker in $C^-$ ($C^+ > C^-$), which suggested that the creation of abstract contextual information depended on the initial perception of context stimuli. Structure 3 appeared mainly in the β band (10–30 Hz), and contained primarily bottom-up connections from the posterior temporal cortex (mainly the area TEO) to the anterior temporal cortex (mainly the temporal pole) and the lateral and medial PFC.

Structure 4 showed the same generalized context dependence as Structure 3, but during the *Response* period when context stimuli were absent and only in $C^+R_f$ (not in $C^+R_n$ and $C^-$). The absence of context dependence in $C^+R_n$ and $C^-$ suggested that Structure 4 required both vM responses with high emotional valence and its context. Moreover, Structure 4 exhibited spatial and spectral characteristics similar to Structure 3 (*Figure 5—figure supplement 1*). We conclude that Structures 3 and 4 represent the same or very similar neural substrate, differing only in when and how they were activated. Structure 3 corresponds to the initial formation/encoding of the contextual information, while Structure 4 represents the $R_f$-triggered reactivation/retrieval of the contextual information. Therefore, Structures 3 and 4 represent the generalized, abstract perceptual and cognitive content of the context.

Structure 5 showed context dependence ($C_m \approx C_h > C_w$) in $C^+R_f$ (not in $C^+R_n$ and $C^-$), and response dependence ($R_f > R_n$) in $C^+$ (not in $C^-$) during the *Response* period, and appeared mainly in α and low-β

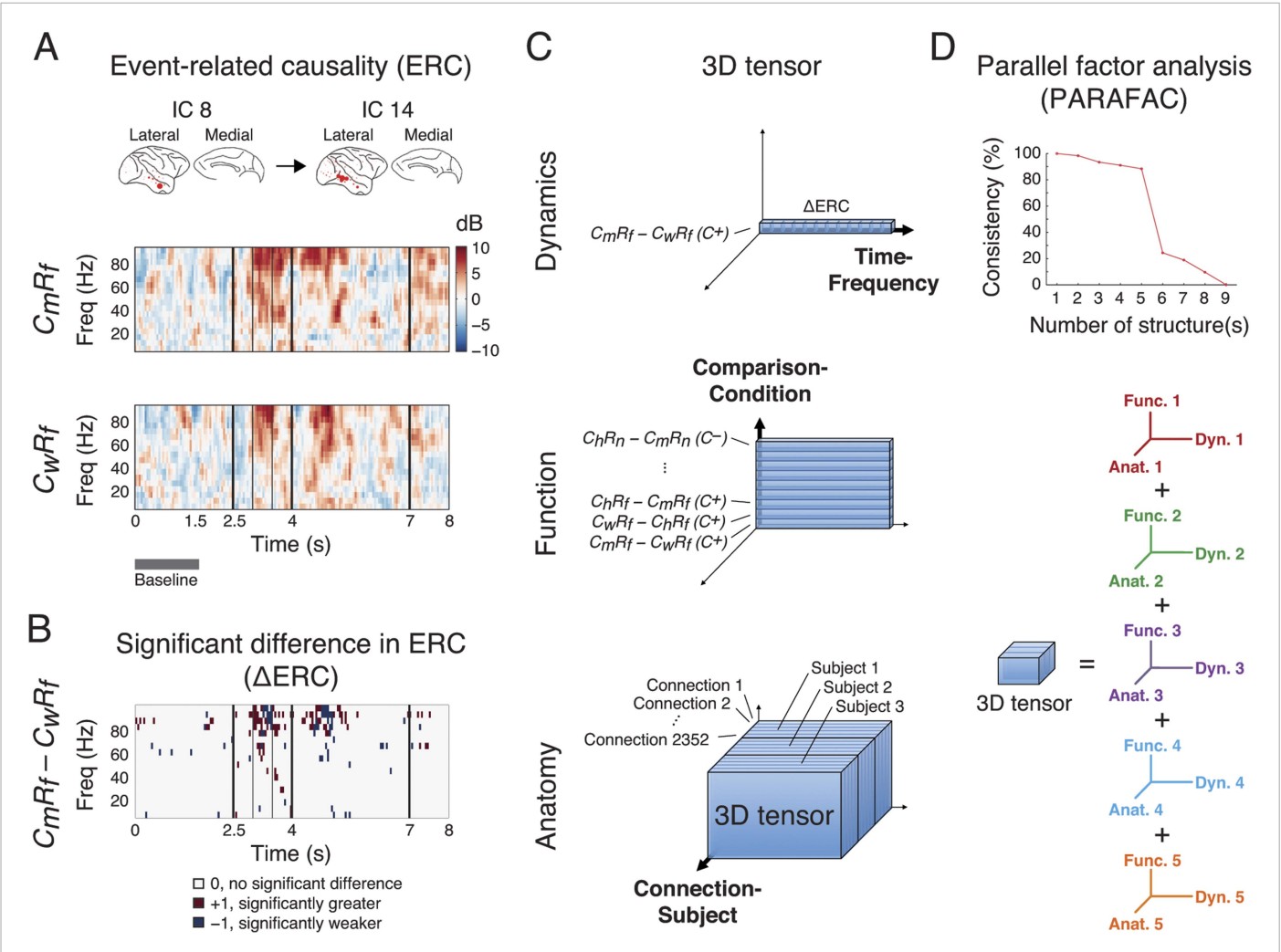

**Figure 3**. Identification of latent structures in context- and response-dependent cortical network interactions. (**A**) Event-related causalities (ERCs) between cortical areas. Example ERCs for a connection (IC 8 to IC 14, the corresponding cortical areas shown on the top) in two scenarios ($C_mR_f$ and $C_wR_f$ in $C^+$) from Subject 1 are shown. Each ERC represents the spectro-temporal dynamics of causality evoked by a scenario, calculated as the logarithmic ratio between the direct directed transfer function (dDTF) and corresponding baseline values (baseline: gray bar), and measured in decibel (dB). Black vertical lines represent task events explained in *Figure 2*. (**B**) ΔERCs, or the significant differences in ERCs between the two trial types ($C_mR_f - C_wR_f$) ($\alpha_{FDR} = 0.05$, false discovery rate correction) are shown. The results were either 0 (no significant difference), +1 (significantly greater), or −1 (significantly weaker). (**C**) 3D tensor of ΔERCs. The data for the entire study were organized in three dimensions: dynamics (top), function (middle), and anatomy (bottom). *Top*: ΔERCs shown in **B** describe the dynamics of difference in causality of a connection between two trial types, presented as a vector in 3D space (illustrated as a bar, where each segment represents a ΔERC value). *Middle*: For the same connection, ΔERCs from other comparisons were pooled to describe the functional dynamics of the connection (illustrated as a plate). *Bottom*: Functional dynamics from all connections were pooled to summarize the functional network dynamics in a subject (illustrated as a block). The data from all subjects were further combined to assess common functional network dynamics across subjects. (**D**) Parallel factor analysis (PARAFAC) extracted five dominant structures from the 3D tensor with consistency (>80%, also see *Figure 3—figure supplement 2*). Each structure represented a unique pattern of network function, dynamics, and anatomy (e.g., Func. 1, Dyn. 1, and Anat. 1 for Structure 1).

The following figure supplements are available for figure 3:

**Figure supplement 1**. Independent component analysis (ICA) results from 3 subjects.

**Figure supplement 2**. PARAFAC revealed five dominant structures in the 3D tensor.

**Figure supplement 3**. Five latent network structures were robust against ICA model order selection.

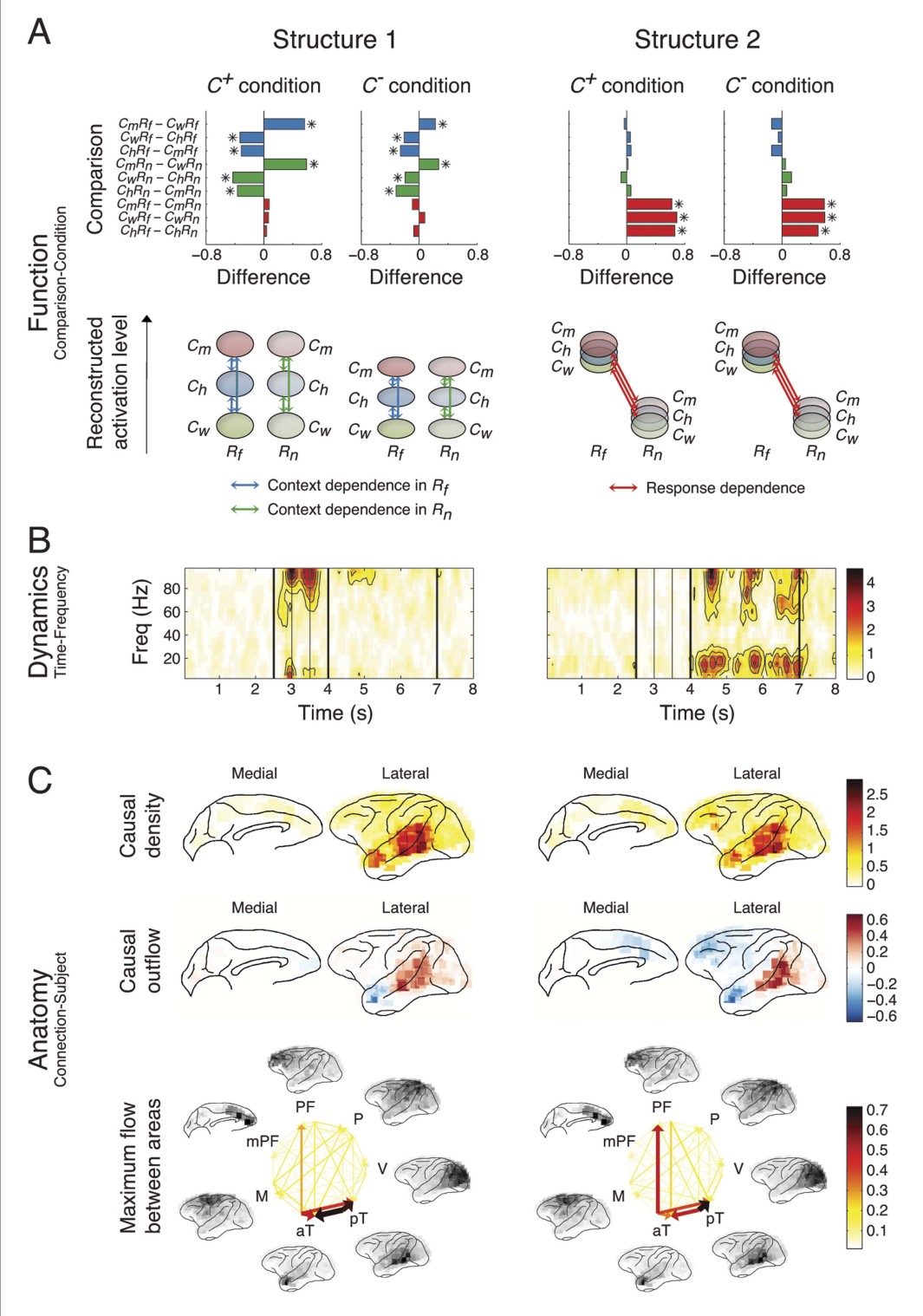

**Figure 4**. Network structures for perception of context and response. Each structure was defined by three dimensions: function, dynamics, and anatomy. (**A**) *Function*: The function dimension showed each structure's context and response dependence. *Top*: For each structure, the first tensor dimension contained 18 differences for nine pairwise comparisons in the $C^+$ or $C^-$ condition. Significant differences are highlighted (*, $\alpha = 0.05$, see the 'Materials and methods'). *Bottom*: The comparisons with significant differences were used to reconstruct how each structure was selectively activated. Each oval and its vertical position represent the trial type and its activation level, *Figure 4. continued on next page*

*Figure 4. Continued*

respectively. Blue, green, or red arrows indicate significant context dependence under $R_f$, significant context dependence under $R_n$, and significant response dependence, respectively (each corresponds to a significance highlighted in the top panel). (**B**) *Dynamics*: The dynamics dimension indexed each structure's activation in different times and frequencies. Black vertical lines represent events, as explained in *Figure 2*. (**C**) *Anatomy*: The anatomy dimension showed each structure's activation in different connections. Three connectivity statistics, averaged across subjects after brain map registration, are shown on the lateral and medial cortices. *Top*: Cortical areas with greater causal density represent areas with busier interactions. *Middle*: Cortical areas with positive (red) and negative (blue) causal outflows represent the sources and sinks of interactions, respectively. *Bottom*: The direction and strength of each maximum flow between areas are indicated by the direction and size (and color) of an arrow, respectively. Seven cortical areas were determined for visualization: the visual (V), parietal (P), prefrontal (PF), medial prefrontal (mPF), motor (M), anterior temporal (aT), and posterior temporal (pT) cortices.

The following figure supplements are available for figure 4:

**Figure supplement 1**. Causal density in individual subjects.

**Figure supplement 2**. Causal outflow in individual subjects.

**Figure supplement 3**. Maximum flow between areas in individual subjects.

**Figure supplement 4**. Robust connectivity across subjects.

---

bands (5–20 Hz). Anatomically, the structure showed primarily top-down connections between posterior temporal cortex, the anterior temporal cortex, and the lateral and medial PFC. Remarkably, Structure 5 is the only one demonstrating clear top-down connections, with the same context and response dependence as the gaze behavior (see *Figure 2B*). These results suggest that Structure 5 corresponds to a network for the context-dependent feedback modulation of eye gaze or visual attention during the task, and the other four structures index internal processes that lead to this behavioral modulation.

## Functional, dynamical, and anatomical correlations between network structures

We investigated how the structures coordinated with each other during the task by examining how they correlated with each other in the functional, dynamical, and anatomical domains.

To study function, we evaluated how each structure's context and response dependence correlated with others', by measuring correlation coefficients of structures' differences in comparisons across contexts in $R_f$, across contexts in $R_n$, and across responses (*Figure 6A*) (detailed in the 'Materials and methods'). Significant correlations between two structures indicated that one structure's activation affected another's, and vice versa, demonstrating a causal interdependence or a common external driver. Across contexts in $R_f$ (*Figure 6A*, left), Structure 1 significantly correlated to Structures 3 and 4, which were themselves significantly correlated to Structure 5. However, across contexts in $R_n$ (*Figure 6A*, middle), a significant correlation was found only between Structures 1 and 3. These results confirmed that sensory perception of the context stimuli could be significantly correlated to the formation of an abstract context, and, in turn this abstract context could be significantly correlated to its reactivation and top-down modulation when a response had high emotional valence. Across responses (*Figure 6A*, right), Structure 2 significantly correlated to Structure 4, which was itself significantly correlated to Structure 5. This indicated that that top-down modulation is the integration of response information and abstract context information.

To examine dynamics, we tested whether network structures had mutually exclusive or overlapping spectro-temporal dynamics. We measured the temporal dynamics of each structure by summing up the activation in the second tensor dimension across frequencies. Significant correlations in temporal dynamics were found between structures activated in the *Context* period (Structures 1 and 3) and the *Response* period (Structures 2, 4, and 5) (*Figure 6B*, left). We then measured the spectral profile of each structure by summing up the activation in the second tensor dimension over time. Significant

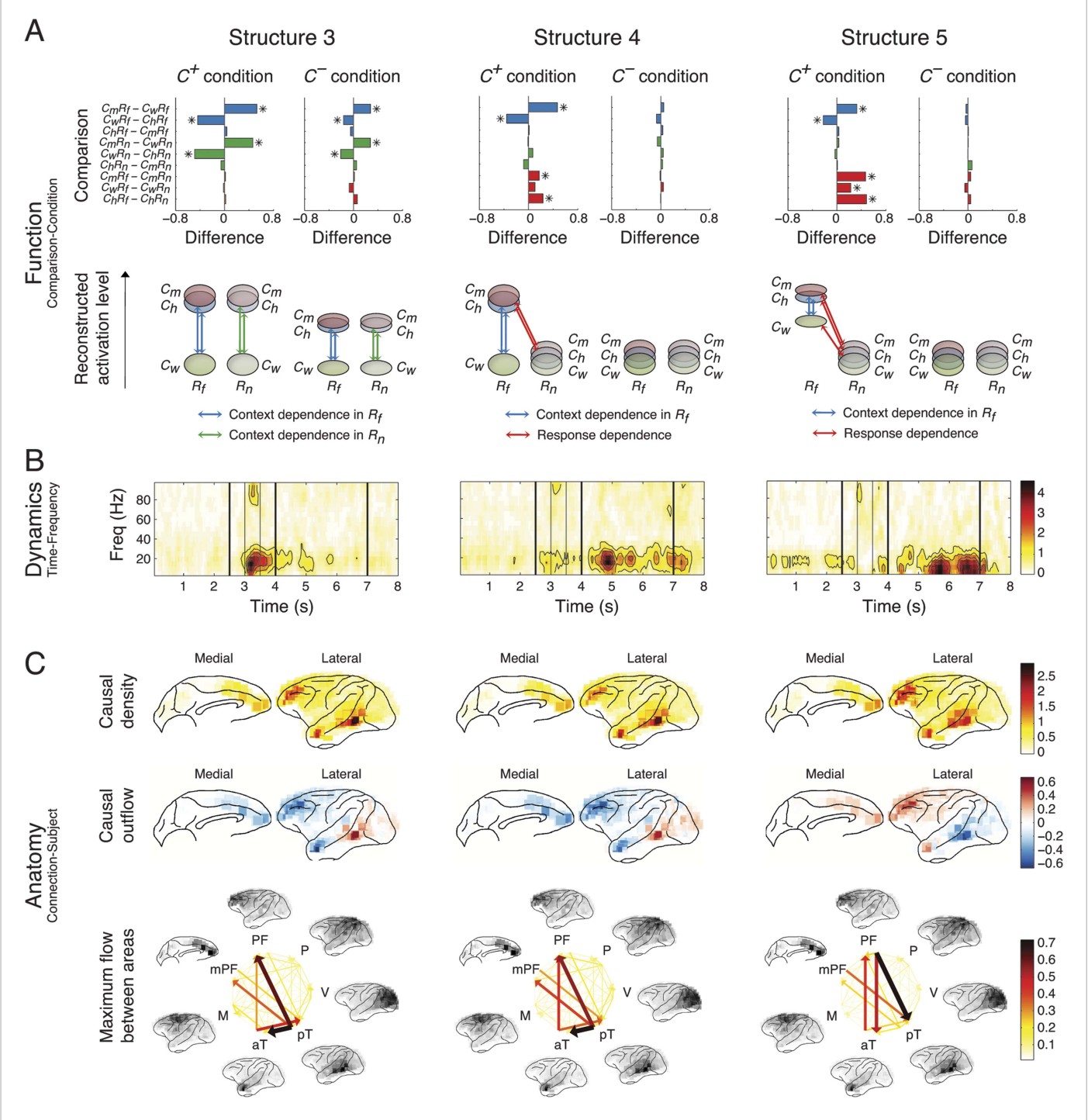

**Figure 5**. Network structures for context representation and modulation. The function (**A**), dynamics (**B**), and anatomy (**C**) dimensions of Structures 3, 4, and 5. Structures 3 and 4 represent the initial formation/encoding and later reactivation/retrieval of abstract context information, respectively, and Structure 5 represents context-dependent top-down feedback that modulates eye gaze or visual attention. Same presentation details as in *Figure 4*.

The following figure supplement is available for figure 5:

**Figure supplement 1**. Spatial and spectral characteristics of network structures.

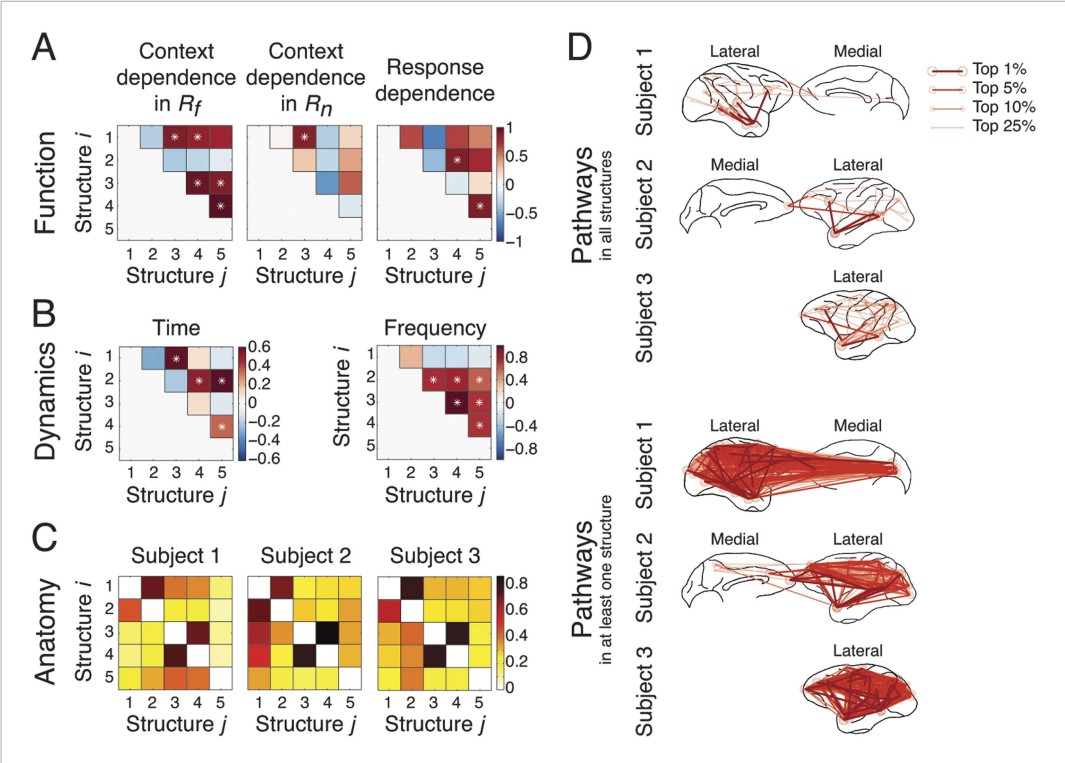

**Figure 6**. Coordination and co-activation of network structures. (**A**) *Functional coordination*: The coordination between structures was evaluated by the correlation coefficients between structures' context and response dependence (the differences shown in *Figures 4A, 5A*). Each panel illustrates how Structure *i* (y-axis) correlated with Structure *j* (x-axis) in context dependence in $R_f$ (left), context dependence in $R_n$ (middle), and response dependence (right). Significant correlations are indicated as asterisks ($\alpha = 0.05$) (see 'Materials and methods'). (**B**) *Dynamic co-activation*: The dynamics correlation was shown by correlation coefficients between structures' temporal and spectral activation. Each panel shows how Structure *i* correlated with Structure *j* in temporal dynamics (left) and frequency profile (right). Significant correlations are indicated as asterisks ($\alpha = 0.05$). (**C**) *Anatomical overlap*: The anatomical similarity was indexed by the ratio of shared anatomical connections between structures. Each panel illustrates the ratio of the number of shared connections between Structures *i* and *j* and the total number of connections in Structure *i*. Results obtained from three subjects are shown separately. (**D**) Undirected pathways of connections shared by all structures for each subject (top), and those appearing in at least one structure for each subject (bottom). The lateral cortical surface is shown on the left for Subject 1, and on the right for Subjects 2 and 3. Shared pathways (lines) between two cortical areas (circles) of the top 1, 5, 10, and 25% connections are shown. Pathways with greater strengths are overlaid on those with weaker strengths.

correlations in spectral profiles were found among structures with β band activation (Structures 2, 3, 4, and 5) (*Figure 6B*, right).

To investigate anatomy, we identified directed connections with the top 10% strengths in the third tensor dimension, and examined the shared top 10% connections between structures for each subject (*Figure 6C*). The numbers of shared connections between all structures were particularly high (>70% shared) between Structures 3 and 4 (abstract contextual information), and Structures 1 and 2 (perception). We examined the undirected pathways that exclude the directionality of connections and found pathways shared by all structures and subjects in and from the temporal cortex to PFC (*Figure 6D*, top). Pathways appearing in at least one structure were widespread across cortex (*Figure 6D*, bottom).

These results demonstrate the functional coordination and spatio-spectro-temporal co-activation of the five identified network structures, and reveal the multiplexing property of large-scale neuronal interactions in brain: simultaneous information transfer in similar frequency bands along similar anatomical pathways could be functionally reconstituted into distinct cognitive operations depending on other networks' ongoing status. This type of information would be difficult to extract from traditional EEG/MEG/fMRI analyses.

## Discussion

In this study, we demonstrate that context can be represented by dynamic communication structures involving distributed brain areas and coordinated within large-scale neuronal networks, or neurocognitive networks (*Varela et al., 2001*; *Fries, 2005*; *Bressler and Menon, 2010*; *Siegel et al., 2012*). Our analysis combines three critical properties of neurophysiology—function, dynamics, and anatomy—to provide a high-resolution large-scale description of brain network dynamics for context. The five network structures we identified reveal how contextual information can be encoded and retrieved to modulate behavior with different bottom-up or top-down configurations. The coordination of distributed brain areas explains how context can regulate diverse neurocognitive operations for behavioral flexibility.

### Context is encoded by interactions of large-scale network structures

These findings show that context can be encoded in large-scale bottom-up interactions from the posterior temporal cortex to the anterior temporal cortex and the lateral and medial PFC. The PFC is an important node in the 'context' network (*Miller and Cohen, 2001*; *Bar, 2004*), where the lateral PFC is believed to be critical for establishing contingencies between contextually related events (*Fuster et al., 2000*; *Koechlin et al., 2003*), and the medial PFC is involved in context-dependent cognition (*Shidara and Richmond, 2002*) and conditioning (*Fuster et al., 2000*; *Koechlin et al., 2003*; *Frankland et al., 2004*; *Quinn et al., 2008*; *Maren et al., 2013*). Our results indicate that abstract contextual information can be encoded not only within the PFC, but in PFC interactions with lower-level perceptual areas in the temporal cortex. These dynamic interactions between unimodal sensory and multimodal association areas could explain the neuronal basis of why context networks can affect a wide range of cognitive processes, from lower-level perception to higher-level executive functions.

Apart from the bottom-up network structure that encodes abstract context, we discovered other network structures that process either lower-level sensory inputs for context encoding or integrate contextual information for behavioral modulation. Evidently, brain contextual processing, from initial perception to subsequent retrieval, is represented not by sequential activation but rather sequential modular communication among participating brain areas. Thus, we believe that the network structures we observed represent a module of modules, or 'meta-module' for brain communication connectivity. Further investigation of this meta-structure organization for brain network communication could help determine how deficits in context processing in psychiatric disorders such as schizophrenia (*Barch et al., 2003*) and post-traumatic stress disorder (*Milad et al., 2009*) could contribute to their etiology.

### Cognition as a modular organization consisting of network structures

These results suggest a basic structural organization of large-scale communication within brain networks that coordinate context processing, and provide insight into how apparently seamless cognition is constructed from these network communication modules. In contrast to previous studies where brain modularity is defined as a 'community' of spatial connections (*Bullmore and Sporns, 2009*; *Sporns, 2011*), or coherent oscillations among neuronal populations in overlapping frequency bands (*Siegel et al., 2012*), our findings provide an even more general yet finer grained definition of modularity based on not only anatomical and spectral properties, but also temporal, functional, and directional connectivity data. The relationships among network properties in the functional, temporal, spectral, and anatomical domains revealed network structures whose activity coordinated with each other in a deterministic manner (*Figure 7A*), despite being highly overlapping in time, frequency, and space (*Figure 7B*). Such multiplexed, yet large-scale, neuronal network structures could represent a novel meta-structure organization for brain network communication. Further studies will be needed to show whether these structures are components of cognition.

### Applications for large-scale functional brain network mapping

We developed an analytical approach using an unbiased deconvolution of comprehensive network activity under well-controlled and staged behavioral task conditions. This workflow enabled us to identify novel network structures and their dynamic evolution during ongoing behavior. In principle, this approach can be generally useful to investigate how network structures link neural activity and behavior. However, we caution that the latent network structures we identified were extracted computationally, and therefore will require further confirmatory experiments to verify their biological significance, particularly the causality of the connectivity patterns within each structure

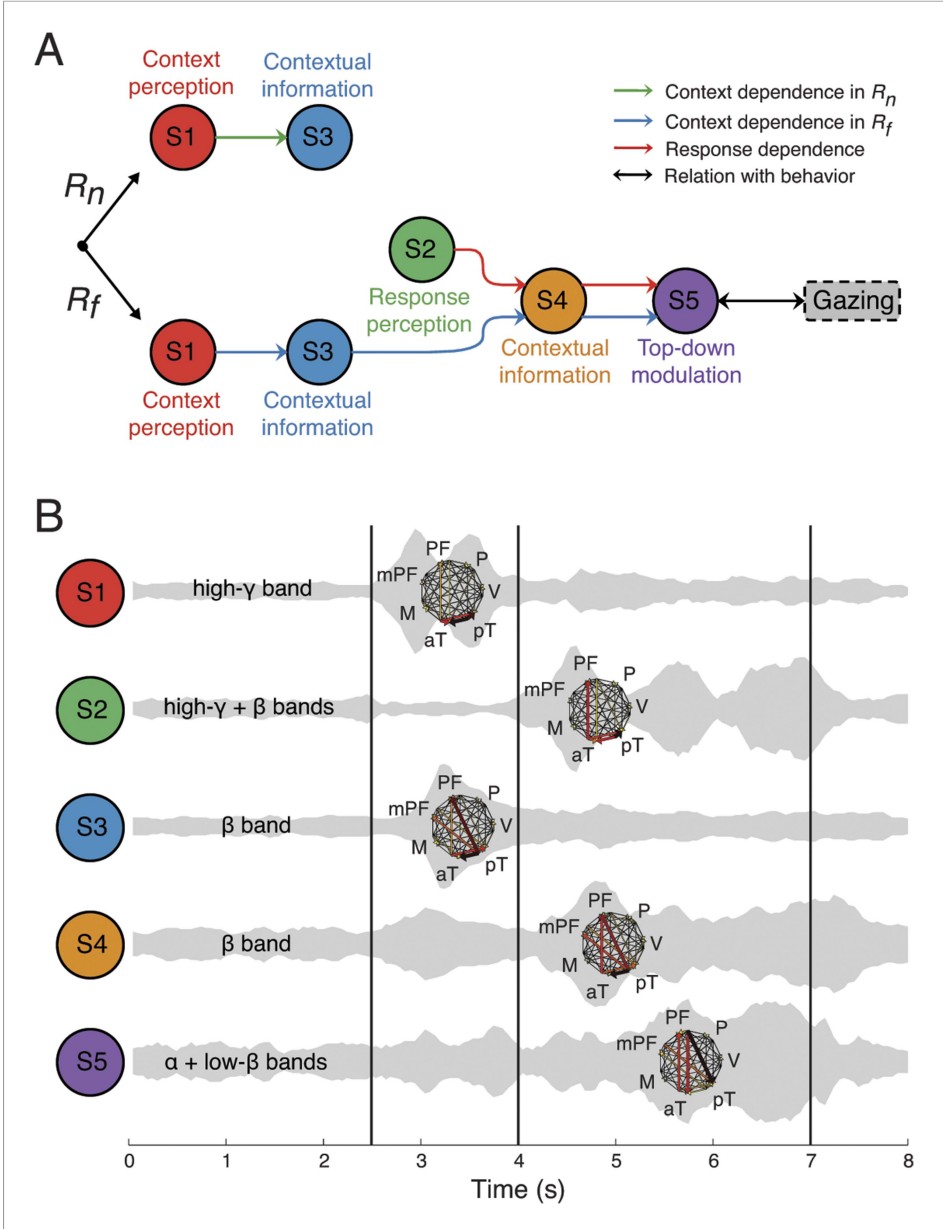

**Figure 7.** Context as a sequence of interactions between network structures. (**A**) Coordination between network structures (S1 to S5, circles), under $R_n$ (top) or $R_f$ (bottom) responses. In both response contingencies, context perception (S1) encoded contextual information (S3). However, when the response stimulus contained high emotional valence ($R_f$, bottom), response perception (S2) reactivates the contextual information (S4), resulting in top-down modulation feedback (S5) that shares the same context and response dependence as the gazing behavior (black arrow and rounded rectangles). Green, blue, and red arrows represent correlations in context dependence in $R_n$, context dependence in $R_f$, and in response dependence, respectively (see **Figure 6A**). (**B**) Temporal, spectral, and spatial profiles and overlap in defined network structures. Network structures can be characterized by frequency range (labeled on the left) and connectivity pattern (shown on the right). Their temporal activations are plotted over trial time, with a 'sound-like' presentation, where a higher volume represents stronger activation. Black vertical lines represent the events as indicated in **Figure 2**.

and the functional links bridging different structures. The biological meaning of the identified network structures could be achieved by selective manipulation of neuronal pathways by electrical or optogenetic stimulation linked to the ECoG array by neurofeedback, or neuropharmacological manipulations.

The general class of network structures we identified are not necessarily unique to context, By recording with a hemisphere-wide ECoG array and applying our analytical methodologies to other cognitive behaviors and tasks in non-human primates, we fully expect to observe similar network structures. Our approach of pooling large-scale data across subjects may be useful to extract network structures that are generalizable, because neural processes specific to individual subjects or trials will cancel. Indeed, the stable and consistent trial responses across subjects in our chronic ECoG recordings suggest that the network structures we isolated may be candidate innate, elemental units of brain organization. Conversely, future identification of unique differences in network structures between subjects could offer insight into structures related to individual trait and state variability, and the network-level etiology of brain diseases (*Belmonte et al., 2004*; *Uhlhaas and Singer, 2006*).

## Materials and methods

### Subjects and materials

Customized 128-channel ECoG electrode arrays (Unique Medical, Japan) containing 2.1 mm diameter platinum electrodes (1 mm diameter exposed from a silicone sheet) with an inter-electrode distances of 5 mm were chronically implanted in the subdural space in three Japanese macaques (Subjects 1, 2, and 3). The details of surgical methods can be found on Neurotycho.org. In Subject 1, electrodes were placed to cover most of the lateral surface of the right hemisphere, also the medial parts of the frontal and occipital lobes. In Subject 2, a similar layout was used, but in the left hemisphere. In Subjects 3, all electrodes were placed on the lateral surface of the left hemisphere, and no medial parts were covered. The reference electrode was also placed in the subdural space, and the ground electrode was placed in the epidural space. Electrical cables leading from the ECoG electrodes were connected to Omnetics connectors (Unique Medical) affixed to the skull with an adaptor and titanium screws. The locations of the electrodes were identified by overlaying magnetic resonance imaging scans and x-ray images. For brain map registration, the electrode locations and the brain outlines from Subjects 1 and 3 were manually registered to those from Subject 2 based on 13 markers in the lateral hemisphere and 5 markers in the medial hemisphere (see *Figure 1—figure supplement 1*).

All experimental and surgical procedures were performed in accordance with the experimental protocols (No. H24-2-203(4)) approved by the RIKEN ethics committee and the recommendations of the Weatherall report, 'The use of non-human primates in research'. Implantation surgery was performed under sodium pentobarbital anesthesia, and all efforts were made to minimize suffering. No animal was sacrificed in this study. Overall care was managed by the Division of Research Resource Center at RIKEN Brain Science Institute. The animal was housed in a large individual enclosure with other animals visible in the room, and maintained on a 12:12-hr light:dark cycle. The animal was given food (PS-A; Oriental Yeast Co., Ltd., Tokyo, Japan) and water ad libitum, and also daily fruit/dry treats as a means of enrichment and novelty. The animal was occasionally provided toys in the cage. The in-house veterinary doctor checked the animal and updated daily feedings in order to maintain weight. We have attempted to offer as humane treatment of our subject as possible.

### Task design

During the task, each monkey was seated in a primate-chair with its arms and head gently restrained, while a series of video clips was presented on a monitor (*Videos 1–6*). In one recording session, each of six video clips was presented 50 times, and all 300 stimuli were presented in a pseudorandom order in which the same stimulus would not be successively presented. In order to keep the monkey's attention to the videos, food items were given after every 100 stimuli. Each monkey participated three recording sessions within a week. Each stimulus consisted of three periods: *Waiting*, *Context*, and *Response* periods. During the *Waiting* period, a still picture created by pixel-based averaging and randomizing the all frames of stimuli was presented without sound for 2. During the first 0.5 s of the *Context* period, a still image of an actor (a monkey) and an opponent (a monkey, a human, or wall) was presented with the sound associated with the opponents. The actor was always positioned on the left side of the image. Then a curtain in the video started to close from the right side toward the center to cover the opponent. The curtain closing animation took 0.5 s, and the curtain stayed closed for another 0.5 s. During the *Response* period, one of two emotional expressions of the actor (frightening or neutral) was presented with sound for 3 s, followed by the *Waiting* period of the next trial.

### ECoG and behavior recordings

An iMac personal computer (Apple, USA) was used to present the stimuli on a 24-in LCD monitor (IOData, Japan) located 60 cm away from the subject. The sound was presented through one MA-8BK monitor speaker (Roland, Japan) attached to the PC. The experiments were run by a program developed in MATLAB (MathWorks, USA) with Psychtoolbox-3 extensions (*Brainard, 1997*). The same PC was used to control the experiments and the devices for recording monkey's gaze and neural signals via USB-1208LS data acquisition device (Measurement Computing Co., USA). A custom-made eye-track system was used for monitoring and recording the monkey's left (Subject 1) or right (Subjects 2 and 3) eye at 30 Hz sampling (*Nagasaka et al., 2011*). Cerebus data acquisition systems (Blackrock Microsystems, USA) were used to record ECoG signals with a sampling rate of 1 kHz.

### Trial screening

Trials during which the subject's eye position was within the screen area more than 80% of the time during the first 0.5 s of the *Context* period were classified as $C^+$ trials. The rest of the trials were identified as $C^-$ trials, where the subject either closed its eyes or the eye position was outside the screen or outside the recording range ($\pm30°$).

### Data analysis

#### ICA

ICA was performed on the data combined $C^+$ and $C^-$ trials to acquire a common basis for easier interpretation of the results. On the other hand, dDTF and the following analyses (ERC and SD-ERC) were calculated from $C^+$ and $C^-$ trials separately, and later combined in PARAFAC analysis.

#### Preprocessing

The 50 Hz line noise was removed from raw ECoG data by using the Chronux toolbox (*Bokil et al., 2010*). The data was then downsampled four times, resulted in a sampling rate of 250 Hz. Trials with abnormal spectra were rejected by using an automated algorithm from the EEGLAB library (*Delorme et al., 2011*), which has been suggested as the most effective method for artifact rejection (*Delorme et al., 2007*). The numbers of trials preserved are shown in *Table 1*.

#### Model order selection

The model order, that is, the number of components, for ICA was determined by the PCA of the data covariance matrix, where the number of eigenvalues accounted for 90% of the total observed variance. The resulted model order is shown in *Table 1*.

#### ICA algorithm

ICA was performed in multiple runs using different initial values and different bootstrapped data sets by using the ICASSO package (*Himberg et al., 2004*) with the FastICA algorithm (*Hyvärinen and Oja, 1997*), which could significantly improve the reliability of the results (*Meinecke et al., 2002*). In the end, artifactual components with extreme values and abnormal spectra were discarded by using an automated algorithm from EEGLAB (*Delorme and Makeig, 2004*). The number of components preserved after this screening process is shown in *Table 1*.

#### Multivariate spectral causality

Spectral connectivity measures for multitrial multichannel data, which can be derived from the coefficients of the multivariate autoregressive model, require that each time series be covariance stationary, that is, its mean and variance remain unchanged over time. However, ECoG signals are usually highly nonstationary, exhibiting dramatic and transient fluctuations. A sliding-window method was implemented to segment the signals into sufficiently small windows, and connectivity was calculated within each window, where the signal is *locally* stationary.

#### Preprocessing

Three preprocessing steps were performed to achieve local stationarity: (1) detrending, (2) temporal normalization, and (3) ensemble normalization (*Ding et al., 2000*). Detrending, which is the subtraction of the best-fitting line from each time series, removes the linear drift in the data. Temporal

normalization, which is the subtraction of the mean of each time series and division by the standard deviation, ensures that all variables have equal weights across the trial. These processes were performed on each trial for each channel. Ensemble normalization, which is the pointwise subtraction of the ensemble mean and division by the ensemble standard deviation, targets rich task-relevant information that cannot be inferred from the event-related potential (*Ding et al., 2000*; *Bressler and Seth, 2011*).

## Window length selection

The length and the step size of the sliding-window for segmentation were set as 250 ms and 50 ms, respectively. The window length selection satisfied the general rule that the number of parameters should be <10% of the data samples: to fit a VAR model with model order $p$ on data of $k$ dimensions ($k$ ICs selected from ICA), the following relation needs to be satisfied: $w \geq 10 \times (k2 \times p/n)$, where $w$ and $n$ represent the window length and the number of trials, respectively.

## Model order selection

Model order, which is related to the length of the signal in the past that is relevant to the current observation, was determined by the Akaike information criterion (AIC) (*Akaike, 1974*). In all subjects, a model order of nine samples (equivalent to $9 \times 4 = 36$ ms of history) resulted in minimal AIC and was selected. The selected model order also passed the Kwiatkowski–Phillips–Schmidt–Shin (KPSS) test, thus maintained local stationarity. Furthermore, the VAR model was validated by the whiteness test and the consistency test.

## Spectral connectivity

dDTF (*Korzeniewska et al., 2003*), a VAR-based spectral connectivity measure, was calculated by using the Source Information Flow Toolbox (SIFT) (*Delorme et al., 2011*) together with other libraries, such as Granger Causal Connectivity Analysis (*Seth, 2010*) and Brain-System for Multivariate AutoRegressive Timeseries (*Cui et al., 2008*). A detailed tutorial of VAR-based connectivity measures can be found in the SIFT handbook (http://sccn.ucsd.edu/wiki/SIFT).

## ERC

To calculate ERC at time $t$ and frequency $f$, or ERC($t$, $f$), dDTF at time $t$ and frequency $f$, or dDTF($t$, $f$), was normalized by the median value during the baseline period at frequency $f$, or dDTF$_{baseline}$ ($f$):

$$dDTF_{baseline}(f) = median(dDTF(t \subset baseline, f)),$$

$$ERC(t, f) = 10 \cdot \log_{10}\left(\frac{dDTF(t, f)}{dDTF_{baseline}(f)}\right). \tag{1}$$

## Comparisons for context and response dependencies

Nine comparisons were performed on the ERCs obtained from different social scenarios in $C^+$ and $C^-$ trials separately. To examine context dependency, three comparisons were performed between scenarios with different contexts followed by the frightened response ($C_hR_f - C_mR_f$, $C_mR_f - C_wR_f$, and $C_wR_f - C_hR_f$), and another three comparisons were performed between scenarios with different contexts followed by the neutral response ($C_hR_n - C_mR_n$, $C_mR_n - C_wR_n$, and $C_wR_n - C_hR_n$). To examine response dependency, three comparisons were performed between scenarios with different responses under the same contexts ($C_hR_f - C_hR_n$, $C_mR_f - C_mR_n$, and $C_wR_f - C_wR_n$). False discovery rate (FDR) control was used to correct for multiple comparisons in multiple hypothesis testing, and a threshold of $\alpha_{FDR} = 0.05$ was used.

## PARAFAC

PARAFAC was performed by using the N-way toolbox (*Andersson and Bro, 2000*), with the following constraints: no constraint on the first tensor dimension, and non-negativity on the second and the third tensor dimensions. The non-negativity constraint was introduced mainly for a more simple visualization of the results. The convergence criterion, that is, the relative change in fit for which the algorithm stops, was set to be 1e-6. The initialization method was set to be DTLD (direct trilinear decomposition) or GRAM (generalized rank annihilation method), which was considered the most accurate method (*Cichocki et al., 2009*). Initialization with random orthogonalized values (repeated 100 times, each time with different random values) was also shown for comparison.

## Connectivity statistics

The connectivity statistics used in this study were calculated from the connectivity matrix (a weighted directed relational matrix) from each latent network structure and each subject, by using the Brain Connectivity Toolbox (*Rubinov and Sporns, 2010*).

## Causal density and outflow

Node strength was measured as the sum of weights of links connected to the node (IC). The causal density of each node was measured as the sum of outward and inward link weights (out-strength + in-strength), and the causal outflow of each node was measured as the difference between outward and inward link weights (out-strength—in-strength). For visualization, each measure was spatially weighted by the absolute normalized spatial weights of the corresponding IC. For example, assume the causal density and causal outflow of IC $ic$ is $Density_{ic}$ and $Outflow_{ic}$, respectively, and the spatial weights of IC $ic$ on channel $ch$ is $W_{ic,ch}$, then the spatial distributions of causal density and causal outflow on each channel will be:

$$Density_{ch} \;=\; \sum_{ic} \frac{abs(W_{ic,ch})}{\max(abs(W_{ic,ch}))} \cdot Density_{ic},$$

$$Outflow_{ch} \;=\; \sum_{ic} \frac{abs(W_{ic,ch})}{\max(abs(W_{ic,ch}))} \cdot Outflow_{ic}.$$

(2)

## Maximum flow between areas

Seven cortical areas were first manually determined: the visual (V), parietal (P), prefrontal (PF), medial prefrontal (mPF), motor (M), anterior temporal (aT), and posterior temporal (pT) cortices. The maximum link among all links connected two areas was selected to represent the maximum flow between the two areas.

## Activation levels from comparison loadings

For each experimental condition ($C^+$ or $C^-$ trials), we determined the significant comparisons for each latent network structure by performing trial shuffling. For each shuffle, dDTF and SR-ERC were recalculated after trial type was randomly shuffled. A new tensor was formed and we estimated how the five spectrotemporal connectivity structures identified from the original data contributed in the shuffled data, by performing PARAFAC on the tensor from the shuffled data with the last two tensor dimensions (Time-Frequency and Connection-Subject) fixed with the values acquired from the original data. Trial shuffling was performed 50 times. The loading from the original data that was significantly different than the loadings from the shuffled data is identified as the significant loading ($\alpha = 0.05$). The comparisons with significant loadings showed the context and response dependencies of each structure, and further revealed the activation levels of each structure in different scenarios.

## Interdependencies among latent network structures

We determined the interdependencies among latent network structures by examining how the activation of one structure could affect the activation of the others. To achieve this, we examined the comparison loadings in the first tensor dimension. For the correlations of structures' activation differences across contexts under $R_f$, we focused on the six comparison loadings representing activation differences across contexts under $R_f$ ($C_hR_f - C_mR_f$, $C_mR_f - C_wR_f$, and $C_wR_f - C_hR_f$) from $C^+$ and $C^-$ trials, and evaluated the correlations of these comparison loadings from different structures. The p-values for testing the hypothesis of no correlation were then computed. For the correlations of structures' activation differences across contexts under $R_n$, we used the same approach to evaluate the correlations of structures' activation differences across contexts under $R_n$ ($C_hR_n - C_mR_n$, $C_mR_n - C_wR_n$, and $C_wR_n - C_hR_n$). For the correlations of structures' activation differences across responses, we evaluated the correlations of structures' activation differences across responses ($C_hR_f - C_hR_n$, $C_mR_f - C_mR_n$, and $C_wR_f - C_wR_n$).

## Acknowledgements

We thank Charles Yokoyama for valuable discussion and paper editing, and Naomi Hasegawa and Tomonori Notoya for medical and technical assistance. We also thank Jun Tani and Douglas Bakkum for their critical comments.

# Additional information

## Funding

| Funder | Grant reference | Author |
|---|---|---|
| Ministry of Education, Culture, Sports, Science, and Technology | Grant-in-Aid for Scientific Research on Innovative Areas, 23118002 | Naotaka Fujii |

The funder had no role in study design, data collection and interpretation, or the decision to submit the work for publication.

## Author contributions

ZCC, Conception and design, Acquisition of data, Analysis and interpretation of data, Drafting or revising the article; YN, Conception and design, Acquisition of data, Drafting or revising the article; NF, Conception and design, Drafting or revising the article, Contributed unpublished essential data or reagents

## Ethics

Animal experimentation: All experimental and surgical procedures were performed in accordance with the experimental protocols (No. H24-2-203(4)) approved by the RIKEN ethics committee and the recommendations of the Weatherall report, 'The use of non-human primates in research'. Implantation surgery was performed under sodium pentobarbital anesthesia, and all efforts were made to minimize suffering. No animal was sacrificed in this study. Overall care was managed by the Division of Research Resource Center at RIKEN Brain Science Institute. The animal was housed in a large individual enclosure with other animals visible in the room, and maintained on a 12:12-hr light: dark cycle. The animal was given food (PS-A; Oriental Yeast Co., Ltd., Tokyo, Japan) and water ad libitum, and also daily fruit/dry treats as a means of enrichment and novelty. The animal was occasionally provided toys in the cage. The in-house veterinary doctor checked the animal and updated daily feedings in order to maintain weight. We have attempted to offer as humane treatment of our subject as possible.

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
