## [Decision Letter]

Thank you for sending your work entitled “Mesoscopic brain networks regulate cognitive enchainment in social monitoring” for consideration at *eLife*. Your article has been favorably evaluated by three reviewers, one of whom, Timothy Behrens (Senior Editor and Reviewing Editor) is a member of our board.

The editor and the other reviewers discussed their comments before we reached this decision, and the editor has assembled the following comments to help you prepare a revised submission.

All three reviewers were very impressed by the unusual nature of the data, by the sophisticated and revealing analysis that was able to simplify complex data to the dynamics of network activity that underlies social observations in the task, and all reviewers were excited by the ability to study this particular brain network in macaque monkeys, given its importance in social cognition in many human studies.

Example praise for the manuscript in review was as follows:

Reviewer 1:

The neural mechanisms mediating social behavior appear to be distributed across the brain, including areas thought to be specialized for social information processing (such as those in the temporal lobe and medial prefrontal cortex) and others that serve more general purpose functions (such as those involved in reward, decision-making, executive control, and attention). One impediment to understanding how these circuits interact to translate sensory perception into behavior is that the methods typically applied suffer from either poor temporal resolution (fMRI in humans), poor spatial resolution (EEG), or limited coverage (single unit recording in animals). The current paper uses wide-scale recordings from intracranial EEG (ECoG) system to simultaneously assess interactions amongst cortical areas during social information processing. Importantly, the authors apply sophisticated, data-driven analytical tools to derive the information flow between and amongst these areas. This is an important advance.

Reviewer 2:

This paper describes a novel approach to extract information about the functional connectivity of distributed brain networks, and how these connections evolve through time as a function of carrying out social observation. The paper is unique in two respects. First, the dataset contains whole-hemisphere ECoG is collected whilst three non-human primates perform a social task (although the authors have published several papers with similar ECoG data previously). Second, the analysis approach is novel and innovative. dDTF is used to identify connections between regions, and then factor analysis is used to isolate how these vary as a function of different task conditions. Using this approach, the authors identify five separate networks (‘structures’). These superficially have similarities in terms of their connectional structure (e.g. structures 1/2, and structures 3/4), but differ in terms of their temporal dynamics and their activation across conditions. Intriguingly, some of the networks contain classic ‘social’ regions, such as the superior temporal sulcus. By examining how these networks evolve through time, the authors attempt to reveal the chain of events underlying different forms of social observation.

Reviewer 3:

This paper reports extremely unusual data from ECoG recordings of macaque monkeys viewing other monkeys engaged in socially threatening situations. It also reports a novel and potentially powerful set of analysis tools for analysing functional networks acquired at high temporal resolution in ECoG data.

There are several key strengths and novelties about the paper.

(1) Intriguing patterns of brain activity and functional connections are reported that have perhaps never been recorded from outside and fMRI scanner, and certainly not in social situations. These patterns are reminiscent of human brain areas that respond to complex social tasks.

(2) The broad coverage of the ECoG data by comparison to most other macaque monkey recordings allows the analysis of information flow between brain areas. Because of the temporal resolution, these analyses can also begin to make directional inferences.

(3) The complex nature of ECoG data requires sophisticated data compression techniques. The authors are extremely inventive in how they analyse their data – they develop tools which compress the data into its digestible patterns, but which maintain the key comparisons between conditions, between brain areas, and between task times. I find this very impressive.

However, there were several features of the data that limited the reviewers' enthusiasm. In brief, these were broadly to do with the task and the over-interpretation of the data. We believe that with a thorough rewrite of the manuscript, to focus on describing the data clearly rather than making interpretations of the data, both in terms of its relevance to particular social behaviours, and in terms of causal mechanisms that are not supported directly by the data, it will be possible to improve the manuscript to remove these concerns.

Specifically:

Comments about the task:

Reviewer #1:

1) The task, which the authors refer to as monitoring of a social context, involves only passive viewing without any differential response expected (or found) on the part of the observer monkey in reaction to different ‘social scenarios’. The only statistically significant difference in behavior is the reported difference in left vs. right gaze positions during the response phase of the trial when monkeys view a frightened vM vs. a neutral vM. Since the authors did not find this result to vary under different ‘social’ contexts, it remains unclear whether/what the observer monkeys made out of the different social situations examined in the study.

2) In the Abstract, it might be more precise to talk about “mapping the network structure” as monkeys viewed scenes leading up to examining the valence on a conspecific's face rather than calling it a “social cognitive behavior”/“social context monitoring” since the behavior per se doesn't inform us in this regard.

3) In the Methods, the task lacks a non-social control, which will in turn depend on what the authors, for the purpose of the expt., define as ‘social’. For instance, is the context of two monkeys looking at each other ‘social’ in which case, the non-social control could be another monkey looking away from the monkey on the left of the screen. Is monkey-monkey looking at each other more ‘social’ than monkey-human looking at each other? These are of course tough questions to answer from a single experiment but it will be still useful to discuss the authors' view on these issues as their basis for designing the expt.

On the other hand, if the element of affect/threatening the monkey on the left is what constitutes social in this case, a non-social control could be a non-threatening/neutral monkey on the right or perhaps an inanimate monkey/human with a threatening expression on it?

Furthermore, the monkey in the video facing an empty wall does not control for the presence of an object or an individual. There are clear sensory/perceptual differences between a monkey face, a human face, and a wall.

Reviewer #3:

It is not clear from the monkey's behaviour that the social nature of the task, rather than the perceptual differences between stimuli, is what is important. In my view this slightly confounds the clear interpretation of these signals as social signals.

Comments about the interpretation of the data:

Reviewer #1:

4) In the Results, the authors say that “subjects tended to focus on the right section during the Response period when the response stimuli were *R*_*f*_ (*C+R*_*f*_ trials)” vs *C+R*_*n*_ trials as well as *C*^*–*^ trials. They conclude that this “indicated that gazing behavior required not only vM responses with high emotional valence, but also the context of vM's response. This suggested that the gazing behavior by the subject represented an automatic or intuitive reaction to socially relatable scenarios (e.g. ‘vM was frightened after being threatened’).”

I think the comparison that the authors analyzed suggest that monkeys are interested in knowing what is behind the curtain when vM is frightened vs. when it is not. This comparison doesn't take into account the nature of context preceding the response since what frightened the monkey – monkey vs. human vs. empty wall – was not compared against each other here. In fact, when the context was indeed taken into account, the authors did not find any significant context dependence at all (Figure 1–figure supplement 3) and hence, in my opinion, the reported difference in gazing location does not make the case for subjects ‘monitoring the context of a social scenario’ at all. This is a critical concern.

Reviewer #2:

I came away with a less than clear impression of what the findings had taught us. I think that this was partly a result of the way the paper was structured. The focus, in the Abstract, Introduction and initial results, was quite heavily on the mathematical technique used as opposed to the results obtained with this technique. Upon reaching the Results, there were some clearly interesting findings, but many of the interpretations were dependent upon a reverse inference from the brain regions activated (Poldrack RA, TiCS 2006), as opposed to an inference based on the task manipulation. I would therefore urge the authors to shift the emphasis in the initial part of the paper towards what their technique and dataset tells us about the dynamics of connectivity in these brain regions, as opposed to being so heavily focused on the methodology.

Reviewer #3:

The nature of the task in combination with the complex analysis often makes interpretation of the results complex, and the authors resort to an interpretation that does not rely on the data. There are examples of this throughout the manuscript. Here is a typical one:

“This result suggests that Structure 5 underlies the context-dependent feedback modulation of response perception linked to social reasoning in the task (‘why vM is frightened?’ or ‘is vM frightened because it was threatened by something?’ or ‘should I be concerned that vM is frightened?’).”

This kind of inference is inappropriate, and is also unnecessary.

The remaining major comments were either about the technicalities of the analysis, or about the reporting of these technicalities. Where possible, we would like you to address these technical concerns. More broadly speaking, we would like you to focus on clarifying the more technical aspects of the manuscript so that the manuscript can be clearly understood by a broad audience.

Other comments:

Reviewer #1:

In the subsection “Deconvolution Analysis or Cortical Information Processing,” wouldn't it be more useful task-wise to identify independent sources by using only *C*^*+*^ trials instead of merging data from *C*^*+*^ & *C*^*–*^ trials for ICA? Did the authors do this analysis? How does it affect the results?

In the first paragraph of the subsection “Dynamic Cognitive Chain Describes Social Context Monitoring” to examine how the network interactions change as a function of context, wouldn't it be better to examine the correlations between activation differences across contexts (*C*_*m*_ & *C*_*h*_ vs. *C*_*w*_) rather than between *C*^*+*^ vs. *C*^*–*^ trials?

In the same subsection, as per Figure 5, none of the correlations with structure 2 are significant. In that case does a causal dependence of structure 4 and 5 on structure 2 apply?

In the second paragraph of the same subsection, does this analysis include the time courses of gaze positions and scanning behavior of all contexts (*C*_*m*_, *C*_*h*_ & *C*_*w*_) and trials (*C*^*+*^ & *C*^*–*^)? Did you find the timing correlations to be different across contexts?

Reviewer #2:

I would therefore urge the authors to shift the emphasis in the initial part of the paper towards what their technique and dataset tells us about the dynamics of connectivity in these brain regions, as opposed to being so heavily focused on the methodology. That said, there were also times where it was unclear what order different techniques were being applied, and how they were being applied. For instance, it is mentioned that ICA was first applied, but it was unclear what the input dimensions of the ICA were, or how the obtained components were subsequently used for the dDTF and PARAFAC analysis. I felt that a clear ‘analysis pipeline’ diagram, starting with raw data and ending with the key results, would be very helpful to include as a supplementary figure.

Finally, I felt that the presentation of the comparison-condition component in Figures 3 and 4 was unclear, in that it seemed diagrammatic whereas presumably it was based upon the statistics of the comparison being performed. A more quantitative approach to presenting this data would help to make it more clear and compelling.

Reviewer #3:

The manuscript is written from a very technical perspective, and does not introduce the key neuroscience issues well. This makes it a very tough read, particularly for a broad interest journal such as *eLife*.

[Editors' note: further revisions were requested prior to acceptance, as described below.]

Thank you for resubmitting your work entitled “Mesoscopic brain networks regulate cognitive enchainment in social monitoring” for further consideration at *eLife*. Your revised article has been evaluated by Timothy Behrens (Senior Editor and Reviewing Editor) and the original reviewers.

The reviews are appended below, and the sentiment in the reviews was reiterated in discussion between reviewers. Essentially, we all remain impressed with the data, are aware that the volume and complexity of data requires innovative new analytical tools, and we still all believe that the proposed analyses are likely very interesting. However, two of the reviewers are clear (and the editorial team agree) that the manuscript cannot possibly be published in a broad interest journal in its present form. Whilst the technical details are more revealing and the claims better substantiated in this current revision, the clarity of the manuscript has, in our view, not improved. It is very difficult indeed to parse the manuscript to understand what the central contribution is. The figures are not well explained in the legends – the legends report details that might be more appropriate in a technical methods section and do not perform the main function of a figure legend, which is to explain how to read the figure.

We would like to ask you to follow the reviewers’ advice below and to restructure and rewrite the paper, so it can be followed in detail by a naïve reader.

Reviewer #1:

I think the authors have done a fine job revising this paper, which presents a novel analytical approach to analyzing density neurophysiological data gathered in monkeys viewing a set of different social scenarios. I'm not yet convinced that the descriptor “social scenario viewing task” is much better than “social monitoring task”– it's both a mouthful and I think still puts too much emphasis on the idea of a task. It might be more concise and precise to say the monkeys were viewing social scenarios.

Reviewer #2:

The authors have put quite some work into making the manuscript clearer, and providing more substantial information concerning the methodology. But the main concerns of the reviewers seem to have held: because of the task design, it is very difficult to draw any strong conclusions about the role of the identified networks in social cognition. In their own words, they acknowledge “without a proper control, we can't conclusively link our results to social cognition.” As such, the authors now explicitly state (in their response) that the main conclusion from the paper is about the development of a novel method. Not much can be learnt about the explicit meaning of the underlying cognitive processes.

The question then becomes, is this novel method of sufficiently broad interest and importance that it will change the views of the community? The authors main claim seems to be that it will reveal previously undiscovered ‘cognitive chains’. What exactly is meant by this? That brain regions are activated sequentially in response to a task, and that different brain areas will be recruited depending upon the cognitive function? On the one hand, this seems to be something that we already know from many years of MEG and EEG research in humans. On the other, it is clear that the spatial resolution of the ECoG data far surpasses this research, and the extraction of network structure is very different from what has gone before. Nevertheless, I still very much struggled to understand whether this extraction of network structure was (a) valid or (b) important. In terms of validity, if I were reviewing this paper at a methods journal, I'd expect a set of examples in simulated data to convince me that the method works well and robustly. In terms of importance, it's precisely because the task is poorly controlled that I don't get an “aha!” moment when examining the results that convinces me that it has definitely worked.

I can see that there is a lot of potential in the paper, and it seems unfair to dismiss what could be an important set of findings about a novel technique. But equally I didn't find the results and structure of the paper sufficiently compelling or clear to warrant publication at present. I'd be open to other reviewers pointing out what they felt was the evidence that the technique works convincingly, or that it has produced a particularly important result.

Reviewer #3:

The authors have done a good job dealing with the technical concerns and no longer over-interpret the data.

However, there still remain concerns about whether the manuscript as currently written can be understood by a broad interest readership, or even a relatively specialised readership within the field. Indeed the mathematical expertise needed even to understand the basic analysis is extreme, and the authors do a very poor job in making the analysis comprehensible. Like the other reviewers, I suspect, I am still finding it difficult to really evaluate the neuroscientific findings, as I cannot fully understand their implications.

Despite the unusual and high quality of the data and the sophisticated nature of the analysis, it is therefore very difficult to understand what we have learnt about the cognitive processes.

For example, the figure legends do not explain how to read the figures. The new diagram asked for by a different reviewer is difficult to understand.

It is essential that the authors address this in both the Abstract and the main text before this can be published in a journal such as *eLife*.

[Editors' note: further revisions were requested prior to acceptance, as described below.]

Thank you for resubmitting your work entitled “Cortical network architecture for context processing in primate brain” for further consideration at *eLife*. Your revised article has been favorably evaluated by Timothy Behrens (Senior Editor).

The new manuscript is, in my view, dramatically clarified, and the data remain extremely exciting. However, in the rewrite the focus of the Abstract and the Introduction has been moved towards the technical achievements of the manuscript. This presents a difficulty for *eLife* because the manuscript has not been reviewed or considered as a technical manuscript. In the Discussion, the nature of the claims is much more balanced between the technical innovations and the neuroscience claims (except perhaps in the first overarching paragraph where the balance is again towards the technical). *eLife* cannot publish the manuscript on the basis of the technical claim alone, but I believe it will be relatively easy to adjust the Abstract and Introduction to highlight and to be clear about the new neuroscience claim.

In my view, the neuroscience claim is still not clearly stated in the Abstract or Introduction. You summarise your findings as follows in the Abstract:

“Collectively, the five structures delineated the flow of information in the network, including two isomorphic variants defining the encoding and retrieval, respectively, of contextual information.”

and in the Introduction:

“The structures we identified provide new insights on how contextual information is processed and help to identify relationships linking network communication and behavior.”

Both of these statements are descriptions of the success of the technique, and not of new neuroscience findings. In brief, what new insights do they provide?

In the Discussion you are much more clear about this in the subsection “Context is Encoded by Interactions of Large-Scale Network Structures”.

If I am correct, it seems like the key claims can be summarised as follows:

a) Large-scale network interactions are different in different contexts.

b) Bottom-up connections from posterior temporal cortex to anterior temporal cortex and medial PFC can encode a context whilst it is being first processed and held on-line.

c) These exact same brain regions exhibit the opposite top-down connectivity only seconds later when the context is being applied to incoming sensory information.

d) The extent to which the bottom-up connectivity is active during the processing of context predicts the extent to which the top-down connectivity will act when the context is later being applied.

To me, these claims seem to be striking and important and it is these claims that the manuscript has been judged on, but these claims no longer appear anywhere in the Abstract or Introduction and are only in a subsection of the Discussion. So I am asking for one further revision. In this round of revision, I am asking for changes to the Abstract, to the Introduction (and possibly also to the Discussion if you choose), which concisely and precisely highlight the new neuroscience claims, and which change the tone of the current version of the manuscript from being largely a methodological innovation, to being a balanced manuscript which introduces a new technique to make a clear and precise claim about the contextual processing of sensory information.

---

## [Author Response]

*Comments about the task*:

Reviewer #1:

*1) The task, which the authors refer to as monitoring of a social context, involves only passive viewing without any differential response expected (or found) on the part of the observer monkey in reaction to different ‘social scenarios’. The only statistically significant difference in behavior is the reported difference in left vs. right gaze positions during the response phase of the trial when monkeys view a frightened vM vs. a neutral vM. Since the authors did not find this result to vary under different ‘social’ contexts, it remains unclear whether/what the observer monkeys made out of the different social situations examined in the study*.

We agree that the observer monkey’s behavior did not directly show “social context monitoring”. Therefore, in the revised manuscript, we have de-emphasized the “social” implication of the conclusions and provided a more precise, conservative interpretation in the paper. Instead, we emphasize the main conclusion that our data reveal a method to simplify complex brain physiological data into mesoscopic network modules underlying implicit cognitive processes related to sensory input of the observer monkey, and whose explicit meaning remains an interesting subject for future investigation. In the revised Introduction we write: “Current methods for studying neurocognitive networks in humans and primates depend on drawing correlations between brain network activity and behavior. […] Here, we employ a new approach that uses an unbiased decomposition of total network activity under diverse task conditions, enabling the computational extraction of latent structure in functional network interactions and post-hoc examination of its dynamic evolution during behavior”.

However, we assert that our task was fundamentally social in nature, since the subjects viewed video clips containing explicit social interactions between a conspecific and second agent. Furthermore, while the eye scanning behavior during the Context period indeed did not show context specificity, they clearly indicated the subjects’ attempt to access the social relationship between the agents and demonstrated the subjects’ observational association with the vM. Therefore, we renamed the task a “social scenario viewing task” to avoid the implication that the task required subjects to process social contextual information. Furthermore, our functional connectivity analysis demonstrated the existence of neural networks that were not observable by external behavior. Thus, we believe that the term “social monitoring” is appropriate since it could refer to an internal process.

*2) In the Abstract, it might be more precise to talk about “mapping the network structure” as monkeys viewed scenes leading up to examining the valence on a conspecific's face rather than calling it a “social cognitive behavior”/“social context monitoring” since the behavior* per se *doesn't inform us in this regard*.

We agree that the observer monkey’s behavior did not show the explicit processing of social context, but we believe there was a clear social observation was in the task (see response above). The reviewer’s proposed description is precise to a fault, but we would like to suggest that “the valence on a conspecific’s face” is part of a social context between two agents that the monkey is observing with interest, as measured by eye scanning between key elements in the scene.

To address this issue, in the Abstract, we now state: “Here, we describe the functional network structure of primate brain during social monitoring where subjects passively viewed social scenarios with staged situational contexts”.

*3) Methods: The task lacks a non-social control, which will in turn depend on what the authors, for the purpose of the expt., define as ‘social’. For instance, is the context of two monkeys looking at each other ‘social’ in which case, the non-social control could be another monkey looking away from the monkey on the left of the screen. Is monkey-monkey looking at each other more ‘social’ than monkey-human looking at each other? These are of course tough questions to answer from a single experiment but it will be still useful to discuss the authors' view on these issues as their basis for designing the expt*.

On the other hand, if the element of affect/ threatening the monkey on the left is what constitutes social in this case, a non-social control could be a non-threatening/ neutral monkey on the right or perhaps an inanimate monkey/ human with a threatening expression on it?

*Furthermore, the monkey in the video facing an empty wall does not control for the presence of an object or an individual. There are clear sensory/perceptual differences between a monkey face, a human face, and a wall*.

The reviewer’s former understanding of the social aspect in the task is correct: the interaction between a conspecific and a second agent in the video. We assert that both monkey-monkey and monkey-human scenarios are social conditions, and that monkey-wall is a non-social condition (no second agent). We agree the monkey-wall condition is not an appropriate non-social control, and without a proper control, we can’t conclusively link our results to social cognition. Therefore, we toned down the social aspect in the revised manuscript (see response above).

In the revised manuscript, we also clarified our rationale for the experimental design: “To allow the controlled recruitment of multiple interdependent cognitive processes in a simple and natural setting, we introduced a social scenario viewing task, where monkey passively viewed video clips in which another monkey (video monkey, or vM) was socially engaged with a second agent.”

Reviewer #3:

*It is not clear from the monkey's behaviour that the social nature of the task, rather than the perceptual differences between stimuli, is what is important. In my view this slightly confounds the clear interpretation of these signals as social signals*.

The subjects’ scanning behavior during the Context period suggested that they were monitoring the relationship between the two agents. Therefore, we believe that the resultant context-dependent brain signals we measured reflect not only perceptual differences, but also social situational differences. However, we agree that further non-social control experiments are needed to definitively conclude this. Thus, in the revised manuscript, we have toned down the implication of social cognition, and instead emphasized the concept of mesoscopic network modules underlying concurrent and intertwined cognitive processes (see response above).

Comments about the interpretation of the data:

Reviewer #1:

*4) Results: The authors say that “subjects tended to focus on the right section during the Response period when the response stimuli were* R_f_
*(*C+R_f_
*trials)” vs* C+R_n_
*trials as well as* C^–^
*trials. They conclude that this “indicated that gazing behavior required not only vM responses with high emotional valence, but also the context of vM's response. This suggested that the gazing behavior by the subject represented an automatic or intuitive reaction to socially relatable scenarios (e.g. ‘vM was frightened after being threatened’)*.*”*

*I think the comparison that the authors analyzed suggest that monkeys are interested in knowing what is behind the curtain when vM is frightened vs. when it is not. This comparison doesn't take into account the nature of context preceding the response since what frightened the monkey – monkey vs. human vs. empty wall – was not compared against each other here. In fact, when the context was indeed taken into account, the authors did not find any significant context dependence at all (Figure 1–figure supplement 3) and hence, in my opinion, the reported difference in gazing location does not make the case for subjects ‘monitoring the context of a social scenario’ at all. This is a critical concern*.

While the eye scanning behavior during the Context period suggested that the subjects were evaluating the social situation presented in the context stimuli, we agree that there’s no contextual specificity in the explicit subjects’ behavior and that the implication of the subjects “monitoring the context of a social scenario” should be presented more conservatively in the text. Therefore, we renamed the task a “social scenario viewing task” instead of “social context monitoring task”, and the cognitive behavior underlying the identified brain networks as “social monitoring” instead of “social context monitoring” (see responses above).

In the revised manuscript, we elaborated on the explanation underlying this more conservative, nuanced conclusion of social monitoring, and instead placed more emphasis on our discovery of brain networks in structured cognition that are not observable via conventional behavioral measurements. In the Abstract we write: “Contextual specificity found in these components was absent in explicit behavioral output, revealing that the connectivity organization in cognition was internally generated.”

Reviewer #2:

*I came away with a less than clear impression of what the findings had taught us. I think that this was partly a result of the way the paper was structured. The focus, in Abstract, Introduction and initial results, was quite heavily on the mathematical technique used as opposed to the results obtained with this technique. Upon reaching the results, there were some clearly interesting findings, but many of the interpretations were dependent upon a reverse inference from the brain regions activated (Poldrack RA, TiCS 2006), as opposed to an inference based on the task manipulation. I would therefore urge the authors to shift the emphasis in the initial part of the paper towards what their technique and dataset tells us about the dynamics of connectivity in these brain regions, as opposed to being so heavily focussed on the methodology*.

In the revised manuscript, we set up our revised emphasis on neurobiology in the Introduction: “Current methods for studying neurocognitive networks in humans and primates depend on drawing correlations between brain network activity and behavior. […] Here, we employ a new approach that uses an unbiased decomposition of total network activity under diverse task conditions, enabling the computational extraction of latent structure in functional network interactions and post-hoc examination of its dynamic evolution during behavior”.

In the revised manuscript, we also clarified the key findings in the Introduction:

“Furthermore, the findings of functionally-specific brain network structures […] in long-hypothesized “cognitive chains”.

Reviewer #3:

*The nature of the task in combination with the complex analysis often makes interpretation of the results complex, and the authors resort to an interpretation that does not rely on the data. There are examples of this throughout the manuscript. Here is a typical one*:

“This result suggests that Structure 5 underlies the context-dependent feedback modulation of response perception linked to social reasoning in the task (‘why vM is frightened?’ or ‘is vM frightened because it was threatened by something?’ or ‘should I be concerned that vM is frightened?’).”

*This kind of inference is inappropriate, and is also unnecessary*.

We agree that our interpretations on the behavior were overreaching, and we removed them across the revised manuscript. In the revised Results section, our interpretations of the network structures now solely rely on the functional specificity (Dimension 1) and temporal structure (Dimension 2), which we elaborated later in the Discussion section based on reverse inference from the frequency profiles (Dimension 2) and the brain regions activated (Dimension 3).

*The remaining major comments were either about the technicalities of the analysis, or about the reporting of these technicalities. Where possible, we would like you to address these technical concerns. More broadly speaking, we would like you to focus on clarifying the more technical aspects of the manuscript so that the manuscript can be clearly understood by a broad audience*.

*Other comments*:

Reviewer #1:

*5) In the subsection “Deconvolution Analysis or Cortical Information Processing,” wouldn't it be more useful task-wise to identify independent sources by using only* C^+^
*trials instead of merging data from* C^+^
*&* C^–^
*trials for ICA? Did the authors do this analysis? How does it affect the results?*

In an earlier analysis presented in the 2012 Japanese Neuroscience Meeting, we screened out the *C*^*–*^ trials and focused only on *C*^*+*^ trials. The results, including the ICA results and the network modules, were very similar to those from the merged *C*^*+*^ and *C*^*–*^ data. Thus, we included *C*^*–*^ trials to increase the variety in experimental conditions (Dimension 1), which could provide additional information to help verify the functionality of each structure. Moreover, our deconvolution of the merged data into discrete networks highlights the strength of our method to analytically simplify complex physiological data, and might be scalable to even larger datasets.

*6) In the first paragraph of the subsection “Dynamic Cognitive Chain Describes Social Context Monitoring” to examine how the network interactions change as a function of context, wouldn't it be better to examine the correlations between activation differences across contexts (*C_m_ & C_h_ vs. C_w_*) rather than between* C^+^
*vs*. *C*^*–*^
*trials?*

Agreed. We re-analyzed the data and revised Figure 5 and the corresponding figure legends and supplemental materials. In the Results, we now state: “To achieve this, we evaluated the correlations of the structures’ activation differences […] top-down modulation can integrate response information with abstract contextual information”.

*In the same subsection, as per*
Figure 5*, none of the correlations with structure 2 are significant. In that case does a causal dependence of structure 4 and 5 on structure 2 apply?*

To address this concern we performed a new analysis (revised Figure 5, right) showing that a significant correlation was observed only between Structures 2 and 5, suggesting that top-down modulation integrated response information with abstract contextual information. Based on the results of this further analysis, we revised the graphical relationships among the 5 structures in Figure 5.

*In the second paragraph of the same subsection, does this analysis include the time courses of gaze positions and scanning behavior of all contexts (*C_m_*,* C_h_
*&* C_w_*) and trials (*C^+^
*&* C^–^*)? Did you find the timing correlations to be different across contexts?*

The analysis included the time courses from all contexts in *C+R*_*f*_. The reason is that only in *C+R*_*f*_ trials did we observe both scanning and gazing behaviors that were significantly different than baseline.

We clarified this in the text: “We focused on the behavior in *C+R*_*f*_ trials, where both scanning and gazing behaviors changed significantly during the trials. We measured cross-correlations between the temporal dynamics of each structure and the median time course of scanning frequency (blue trace in Figure 1) and gaze position (blue trace in Figure 1) (Figure 5).”

Also, we found no significant difference in cross-correlations across contexts. This is not surprising since no significant differences in behavior were observed across contexts (Figure 1*–*figure supplement 4).

Reviewer #2:

*I would therefore urge the authors to shift the emphasis in the initial part of the paper towards what their technique and dataset tells us about the dynamics of connectivity in these brain regions, as opposed to being so heavily focussed on the methodology. That said, there were also times where it was unclear what order different techniques were being applied, and how they were being applied. For instance, it is mentioned that ICA was first applied, but it was unclear what the input dimensions of the ICA were, or how the obtained components were subsequently used for the dDTF and PARAFAC analysis. I felt that a clear ‘analysis pipeline’ diagram, starting with raw data and ending with the key results, would be very helpful to include as a supplementary figure*.

In accord with the reviewers suggestion, we revised the Abstract and Introduction, to better motivate our questions and key findings around how our methodological approach can extract highly concurrent and intertwined cognitive processes that are not be accessible by conventional behavioral measurements (see response above). That is, the neurobiological question we address is fundamentally how to access the structure of physiological information during intrinsic cognition, that at present may be ambiguously defined as mind-wandering or task-free networks.

As suggested by the reviewer, we also created a clear diagram for the analysis pipeline, including the increasing dimensionality of all variables from raw ECoG signals to the final latent network structures (new Figure 2*–*figure supplement 3).

*Finally, I felt that the presentation of the comparison-condition component in*
Figures 3 and 4
*was unclear, in that it seemed diagrammatic whereas presumably it was based upon the statistics of the comparison being performed. A more quantitative approach to presenting this data would help to make it more clear and compelling*.

We agree that it is important to show the original data in the Results, instead of a diagrammatic view. Therefore, in revised Figures 3 and 4, we show the original scores from 18 comparisons and their statistical significance. The original Figures 3 and 4 now are combined in Figure 3—figure supplement 1.

Reviewer #3:

*The manuscript is written from a very technical perspective, and does not introduce the key neuroscience issues well. This makes it a very tough read, particularly for a broad interest journal such as* eLife*.*

In the revised manuscript, we redirected our emphasis to the more accessible issue of how our novel approach can extract functionally meaningful units from high dimensionality data during highly concurrent and intertwined cognitive processes. Importantly, even apparently simple forms of cognition can be complex, showing contingencies and parallel processing, and critical subtleties in the physiological structure of neural data would not be accessible by standard or traditional methods in neuroscience that depend on drawing explicit correlations between brain network activity and an empirical and quantifiable behavior. More generally, we have tried to simplify the text in key areas for the broad *eLife* readership.

*[Editors' note: further revisions were requested prior to acceptance, as described below*.*]*

Reviewer #1:

*I think the authors have done a fine job revising this paper, which presents a novel analytical approach to analyzing how density neurophysiological data gathered in monkeys viewing a set of different social scenarios. I'm not yet convinced that the descriptor “social scenario viewing task” is much better than “social monitoring task” – it's both a mouthful and I think still puts too much emphasis on the idea of a task. It might be more concise and precise to say the monkeys were viewing social scenarios. Other than that concern, I think the paper is ready to go*.

We have significantly restructured the paper. We removed almost all the “social” parts because the social aspect of the task is only between video agents and therefore not central to the main conclusions. Instead, we focus the revision on neural processing of “context”. We study context by combining large-scale recording and analysis to decipher brain networks for fast, internal, concurrent, and interdependent cognitive processing. Our findings demonstrate that context processing is composed of network structures. These can be encoded in bottom-up interactions between sensory and association areas, and top-down interactions for behavioral modulation.

Regarding the task, we now describe it simply as “monkeys watched videos of two agents interacting in different situational contexts.”

Reviewer #2:

*I still remain quite unsure where I stand with this paper. The authors have put quite some work into making the manuscript clearer, and providing more substantial information concerning the methodology. But the main concerns of the reviewers seem to have held: because of the task design, it is very difficult to draw any strong conclusions about the role of the identified networks in social cognition. In their own words, they acknowledge this: “without a proper control, we can't conclusively link our results to social cognition.” As such, the authors now explicitly state (in their response) that the main conclusion from the paper is about the development of a novel method. Not much can be learnt about the explicit meaning of the underlying cognitive processes*.

We agree that it’s difficult to draw firm conclusions on social cognition. Thus, we revised our paper to focus on functional network dynamics for context processing. We show how to extract computational structures from brain network dynamics, and map the internal processing of context. Our findings demonstrate the organization of a “context” network showing context is encoded in large-scale bottom-up interactions among distributed brain areas, and could later be reactivated top down to correlated eye movements.

The link between brain networks and behavior was established by a new eye movement analysis showing context and response dependence in gazing behavior (see new Figure 2). In the previous analysis, we compared eye movements among trials with different contexts (*C*_*h*_*R*_*f*_ vs. *C*_*m*_*R*_*f*_ vs. *C*_*w*_*R*_*f*_, and *C*_*h*_*R*_*n*_ vs. *C*_*m*_*R*_*n*_ vs. *C*_*w*_*R*_*n*_), and found no context dependence. In the new analysis, we performed the same 9 comparisons used in the brain activity analysis to identify context and response dependence: “We performed 9 pairwise comparisons on gaze positions from different context situations, separating C^+^ and C^–^ conditions, to examine their context and response dependence. For context dependence, we compared behaviors from trials with different context stimuli but the same response stimulus (6 comparisons: *C*_*m*_*R*_*f*_ vs *C*_*w*_*R*_*f*_, *C*_*w*_*R*_*f*_ vs *C*_*h*_*R*_*f*_, and *C*_*h*_*R*_*f*_ vs *C*_*m*_*R*_*f*_ for context dependence in *R*_*f*_; *C*_*m*_*R*_*n*_ vs *C*_*w*_*R*_*n*_, *C*_*w*_*R*_*n*_ vs *C*_*h*_*R*_*n*_, and *C*_*h*_*R*_*n*_ vs *C*_*m*_*R*_*n*_ for context dependence in *R*_*n*_). For response dependence, we compared behaviors from trials with the same context stimulus but with different response stimuli (3 comparisons: *C*_*m*_*R*_*f*_ vs *C*_*m*_*R*_*n*_, *C*_*w*_*R*_*f*_ vs *C*_*w*_*R*_*n*_, and *C*_*h*_*R*_*f*_ vs *C*_*h*_*R*_*n*_).”

By using the same comparisons for both behavior and brain activity analyses we could directly compare their context and response dependence.

The question then becomes, is this novel method of sufficiently broad interest and importance that it will change the views of the community?

We believe that our new method and its application to the issue of context processing in primate brain more than meets the high criteria of *eLife* for novelty and advance. The computational methods we use in the paper have literally never been reported before for this type of data. The question of whether it is significant in a biological meaning remains to be verified by others in the field, but we believe this is not a limitation of the study. How to handle big data from neurophysiology and extract biological meaning from its computational analyses is one of if not the most important question in neuroscience in the next 25 years. Our study provides a road map for one type of big data analysis of high density ECoG data.

Our approach of combining large-scale high-resolution neuronal recording and unbiased data-driven analysis is aimed at disentangling simultaneous and interdependent functional brain networks to provide a high-resolution brain-wide network description of context processing. Our findings demonstrate the structure of a “context” brain network showing that contextual information can be encoded in the dynamic interactions between the sensory and association areas, which could explain the neuronal basis of why context processing can affect a wide range of cognitive operations, from lower-level perception to higher-level executive functions. Moreover, our data-driven analysis also discovered other modular brain networks that either process lower-level sensory inputs for context encoding or integrate contextual information for behavioral modulation.

The brain’s ability to process contextual information provides enormous behavioral flexibility, while deficits were thought to lead to psychiatric disorders such as schizophrenia and post-traumatic stress disorder. Even though contextual information processing has been studied in a variety of cognitive domains and brain areas, a comprehensive functional view of the brain circuits that mediate contextual processing and modulation remain lacking. The value of our study is that we can use unbiased computational tools to extract structure from physiological data that appears to track actual cognitive processing. New methods should be evaluated in their potential to provide quantitative assessments of raw data. We do not want to be penalized because our data do not fit with less complex models drawn from fMRI, EEG. MEG methods.

*The authors main claim seems to be that it will reveal previously undiscovered ‘cognitive chains’. What exactly is meant by this? That brain regions are activated sequentially in response to a task, and that different brain areas will be recruited depending upon the cognitive function? On the one hand, this seems to be something that we already know from many years of MEG and EEG research in humans. On the other, it is clear that the spatial resolution of the ECoG data far surpasses this research, and the extraction of network structure is very different from what has gone before*.

In the revision, we have eliminated discussion of “cognitive chains”. The concept we focused on instead is “sequential and modular network interactions”, which means that not only brain regions are activated at different times during a task, but their interactions contain modular structures. In the new Figure 7, we show that modular networks whose activity coordinated with each other in a deterministic manner, even though they were overlapping in time, frequency, and space. We believe that these multiplexed neuronal structures represent a meta-structure organization for brain network communication.

*Nevertheless, I still very much struggled to understand whether this extraction of network structure was (a) valid or (b) important. In terms of validity, if I were reviewing this paper at a methods journal, I'd expect a set of examples in simulated data to convince me that the method works well and robustly. In terms of importance, it's precisely because the task is poorly controlled that I don't get an “aha!” moment when examining the results that convinces me that it has definitely worked*.

I can see that there is a lot of potential in the paper, and it seems unfair to dismiss what could be an important set of findings about a novel technique. But equally I didn't find the results and structure of the paper sufficiently compelling or clear to warrant publication at present. I'd be open to other reviewers pointing out what they felt was the evidence that the technique works convincingly, or that it has produced a particularly important result.

In summary, we have addressed these concerns by:

1) Revising the paper to focus on cortical networks underlying “context processing”, instead of “social cognition” or “cognitive chains”.

2) Providing new behavioral analyses to reveal context-dependent eye movement to enable a more direct comparison between brain activity and behavior.

3) Clarifying and simplifying the methods and results with improved text and figures.

Reviewer #3:

*The authors have done a good job dealing with the technical concerns and no longer over-interpret the data*.

*However, there still remain concerns about whether the manuscript as currently written can be understood by a broad interest readership, or even a relatively specialised readership within the field. Indeed the mathematical expertise needed even to understand the basic analysis is extreme, and the authors do a very poor job in making the analysis comprehensible. Like the other reviewers, I suspect, I am still finding it difficult to really evaluate the neuroscientific findings, as I cannot fully understand their implications*.

Despite the unusual and high quality of the data and the sophisticated nature of the analysis, it is therefore very difficult to understand what we have learnt about the cognitive processes.

*For example, the figure legends do not explain how to read the figures. The new diagram asked for by a different reviewer is difficult to understand*.

*It is essential that the authors address this in both the Abstract and the main text before this can be published in a journal such as* eLife*.*

In this revision, we provided a new figure (Figure 3) to provide a more intuitive idea of our analysis. We also refined the main result figures (Figures 4 and 5), removed unnecessary supplemental figures, and move the more technical descriptions from the main text and figure legends to the Materials and methods. Furthermore, we have given the paper to two readers outside of the field, and incorporated their feedback in this revision.

For neuroscientific findings, we focus the revision on neural processing of “context”: “The findings provide fundamental insights into context processing. […] These dynamic interactions between the unimodal sensory and multimodal association areas could explain the neuronal basis of why context processing can affect a wide range of cognitive operations, from lower-level perception to higher-level executive functions.”

And: “We showed that contextual processing, from initial perception to later modulation, was represented not by sequential activation of functionally specialized brain areas, but by sequential communication among functionally specialized brain areas. Thus, we believe that the network structures we observed represent a module of modules, or “meta-module” for brain connectivity, and further investigation on this meta-structure organization for brain network communication could help determine how deficits in context processing could contribute to psychiatric disorders such as schizophrenia (5) and post-traumatic stress disorder (45).”

*[Editors' note: further revisions were requested prior to acceptance, as described below*.*]*

*[...] If I am correct, it seems like the key claims can be summarised as follows*:

*a) Large-scale network interactions are different in different contexts*.

b) Bottom-up connections from posterior temporal cortex to anterior temporal cortex and medial PFC can encode a context whilst it is being first processed and held on-line.

c) These exact same brain regions exhibit the opposite top-down connectivity only seconds later when the context is being applied to incoming sensory information.

*d) The extent to which the bottom-up connectivity is active during the processing of context predicts the extent to which the top-down connectivity will act when the context is later being applied*.

*To me, these claims seem to be striking and important and it is these claims that the manuscript has been judged on, but these claims no longer appear anywhere in the Abstract or Introduction and are only in a subsection of the Discussion. So I am asking for one further revision. In this round of revision, I am asking for changes to the Abstract, to the Introduction (and possibly also to the Discussion if you choose), which concisely and precisely highlight the new neuroscience claims, and which change the tone of the current version of the manuscript from being largely a methodological innovation, to being a balanced manuscript which introduces a new technique to make a clear and precise claim about the contextual processing of sensory information*.

We deeply appreciate and fully agreed with the comments and suggestions. In this revision, we rewrote the Abstract and significantly restructured the Introduction, with the goal to balance the paper by clarifying and emphasizing the neuroscience insights our paper provides for context processing. We also fine-tuned the Discussion accordingly.